_Resource_

# Integrated multi-omics mapping of mitochondrial dysfunction and substrate preference in Barth syndrome cardiac tissue

Bauke V Schomakers[1,2,12], Adriana S Passadouro[1,3,4,5,12], Maria M Trętowicz [1,4], Pelle J Simpson[6], Yorrick R J Jaspers [1,4], Michel van Weeghel[1,2], Iman Man Hu[1,4], Cathelijne M E Lamboo[1], Denise Cloutier[7], Barry J Byrne [7], Jan Bert van Klinken[1,2,4,8], Paul M L Janssen [9], Sander R Piersma[10], Connie R Jimenez[10], Frédéric M Vaz [1,2,4,11], Gajja S Salomons[1,4], Jolanda van der Velden[3,5], Riekelt H Houtkooper [1,3,4,11✉] & Signe Mosegaard [1,3,4,11✉]

## Abstract

**Barth syndrome (BTHS) is a rare X-linked recessively inherited disorder caused by variants in the TAFAZZIN gene, leading to impaired conversion of monolysocardiolipin (MLCL) into mature cardiolipin (CL). Accumulation of MLCL and CL deficiency are diagnostic markers for BTHS. Clinically, BTHS includes cardiomyopathy, skeletal myopathy, neutropenia, and growth delays. Severely affected patients may require early cardiac transplants due to unpredictable cardiac phenotypes. The pathophysiological mechanisms of BTHS are poorly understood, and treatments remain symptomatic. This study analyzed heart samples from five pediatric male BTHS patients (5 months-15 years) and compared them to tissues from 24 non-failing donors (19–71 years) using an integrated omics method combining metabolomics, lipidomics, and proteomics. The analysis confirmed changes in diagnostic markers (CL and MLCL), severe mitochondrial alterations, metabolic shifts, and elevated heart-failure markers. It also revealed significant interindividual differences among BTHS patients. This study describes a powerful analytical tool for the in-depth analysis of metabolic disorders and a solid foundation for the understanding of BTHS disease phenotypes in cardiac tissues.**

**Keywords** Barth Syndrome; Cardiac Tissue; Integrated Multi-omics; Mitochondrial Dysfunction
**Subject Categories** Cardiovascular System; Chromatin, Transcription & Genomics; Proteomics

## Introduction

Barth syndrome (MIM: 302060) is a rare X-linked recessively inherited mitochondrial disorder caused by pathogenic variants in *TAFAZZIN* (Barth et al, 1983; Bione et al, 1996). The early onset of this disorder includes cardiomyopathy (73% in prenatal period), skeletal muscle myopathies, neutropenia and growth and developmental delays (Taylor et al, 2022; Clarke et al, 2013).

*TAFAZZIN* encodes the Tafazzin protein, a mitochondrial transacylase that is crucial during the cardiolipin (CL) maturation process. CL is a phospholipid of the mitochondrial inner membrane (IM) essential for this membrane's architecture and mitochondrial structure (Houtkooper and Vaz, 2008). After synthesis, CL undergoes remodelling to establish a unique fatty acid composition that is tightly connected to its function in mitochondria. During this process, nascent CL is deacylated to monolysocardiolipin (MLCL) which is then re-acylated by Tafazzin to generate mature CL (Dudek and Maack, 2017). As CL contains four fatty acids and these can all be distinct, there can be many different CL species per tissue or cell type. In the mammalian heart, there is a specific species composition where linoleic acid (18:2) is the predominant fatty acid (Dudek and Maack, 2017). Tafazzin dysfunction leads to the MLCL accumulation and CL depletion since the remodelling process is impaired, making an elevated MLCL/CL ratio a highly specific BTHS diagnostic biomarker (Kulik et al, 2008).

Cardiomyopathy is the most common clinical presentation in infants affected with BTHS and, in the early childhood period, cardiac transplant might be required (Taylor et al, 2022; Li et al, 2021). This could be the case for 12–26% of BTHS individuals as estimated previously (Taylor et al, 2022; Roberts et al, 2012). The

[1]Laboratory Genetic Metabolic Diseases, Amsterdam UMC, University of Amsterdam, Amsterdam, The Netherlands. [2]Core Facility Metabolomics, Amsterdam UMC, University of Amsterdam, Amsterdam, The Netherlands. [3]Amsterdam Cardiovascular Sciences, Amsterdam, The Netherlands. [4]Amsterdam Gastroenterology, Endocrinology and Metabolism, Amsterdam, The Netherlands. [5]Department of Physiology, Amsterdam UMC, Vrije Universiteit Amsterdam, Amsterdam, The Netherlands. [6]Enveda, Boulder, CO, USA. [7]Department of Pediatrics in the College of Medicine, University of Florida, Gainesville, FL, USA. [8]Laboratory for General Clinical Chemistry, Amsterdam UMC, University of Amsterdam, Amsterdam, The Netherlands. [9]Department of Physiology and Cell Biology, College of Medicine, The Ohio State University, Columbus, OH, USA. [10]OncoProteomics Laboratory, Amsterdam UMC, Location VUmc, Medical Oncology, Amsterdam, The Netherlands. [11]Emma Center for Personalized Medicine, Amsterdam, The Netherlands. [12]These authors contributed equally: Bauke V Schomakers, Adriana S Passadouro. ✉E-mail: r.h.houtkooper@amsterdamumc.nl; s.m.nielsen@amsterdamumc.nl

cardiac presentations can be diverse and unpredictable involving ventricular arrhythmias (10–44%), dilated cardiomyopathy (DCM) (96%), hypertrophic cardiomyopathy (HCM) (3%), restrictive cardiomyopathy, left ventricular noncompaction (19–53%), endocardial fibroelastosis and sudden cardiac death (Taylor et al, 2022; Clarke et al, 2013; Spencer et al, 2006; Adès et al, 1993). Standard heart failure (HF) medications have proven to stabilize BTHS patients (Taylor et al, 2022; Clarke et al, 2013). However, after years of cardiac stability, mechanical support and cardiac transplant may still be required despite the risk of a major operation in patients suffering from neutropenia (Zegallai and Hatch, 2021; Hanke et al, 2012; Dedieu et al, 2013). The lack of treatment options and the severity of the phenotype so early in life stimulated scientific research into new possible therapeutic avenues, focusing traditionally on animal models.

Previously, male *TAFAZZIN*-knockdown (KD) mouse models in C57BL/6 genetic background were established and the proteome of cardiac mitochondria isolated from ventricular myocardium was obtained. Proteins related to oxidative phosphorylation (OXPHOS) and ubiquinone biosynthesis were reduced in cardiac mitochondrial. The electron transport chain (ETC) supercomplexes were destabilized and, consequently, the interactions between these supercomplexes and the fatty acid oxidation (FAO) enzymes were abnormal (Huang et al, 2015). When analysing the lipid profile of *TAFAZZIN*-KD mice, a sharp decrease in linoleic acid containing molecular species was identified. Tafazzin knock-down was ultimately associated with alterations in choline and ethanolamine glycerophospholipids and triggered an alteration of myocardial substrate utilization from fatty acids and glucose to amino acid oxidation (Kiebish et al, 2013).

While these animal models have provided novel insights, the translational gap to humans is considerable (Van Norman, 2019). Human studies have mainly focused on sample types not directly involved in cardiac function, such as lymphoblastoid cells (Byeon et al, 2021), plasma (Sandlers et al, 2016) and fibroblasts (Chatzispyrou et al, 2018). Previously, LC-MS/MS was used to establish the lipid profile of lymphoblastoid cell pellets from five BTHS patients (Byeon et al, 2021). Dilysocardiolipin (DLCL), MLCL and CL levels were aberrant, as well as several other lipid species. In a larger cohort, plasma from 23 BTHS patients was analysed using NMR metabolomics and targeted LC-MS metabolomics (Sandlers et al, 2016). Clear differences between BTHS and control samples included acylcarnitines, amino acids, biogenic amines, and glycerophospholipids. Mitochondria were isolated from primary skin fibroblasts of BTHS patients and used for complexome profiling and fluxomics. Partial destabilization of the supercomplexes was detected along with a substrate preference shift towards glutamine. Importantly, it was suggested that further studies were required in order to assess the functional effects these observations could have on high energy tissues, considering that BTHS fibroblasts seemed minimally affected (Chatzispyrou et al, 2018).

In the present study, we have used an integrative omics approach on left ventricular heart tissue samples from five individuals with BTHS for an in-depth analysis of the BTHS cardiac phenotype. The same five cardiac samples were previously analysed using LC-MS/MS to investigate the BTHS cardiac proteome between BTHS cardiomyopathy and idiopathic DCM in children, identifying that long-chain FAO (lcFAO) was impaired

and glucose uptake and utilization was increased (Chatfield et al, 2022). To expand on these findings and map the broad cellular consequences of BTHS, we employed a state-of-the-art multi-omics strategy involving (a) semi-targeted metabolomics, (b) lipidomics of complex lipids, and (c) analytical flow DIA proteomics. All omics methods were performed using a single and unified sample preparation, minimizing sampling variability.

# Results

## Integrative OMICs show phenotypic differences in human BTHS heart samples

In this study, the metabolic, lipidomic and proteomic profiles of five left ventricular heart samples from BTHS individuals and 24 hearts from non-failing donors were examined (Appendix Table S1). The BTHS donors were male, aged between 5 months and 15 years old. Two samples were obtained during autopsy and three at heart transplantation. The Principal Component Analyses (PCA) of metabolomics (123 metabolites, Dataset EV1), lipidomics (1899 lipids, Dataset EV2) and proteomics (1445 proteins, after QC, Dataset EV3) results (Fig. 1A–C) show BTHS triplicates (separately prepared samples from the same donors) individually, allowing for an assessment of their experimental variance. In all three omics analyses, the tight clustering of these triplicates underscores the robustness of the analytical method we present here. The PCA plots show a clear separation of BTHS individuals from the non-failing donors, except for transplantation samples in metabolomics. In each of the omics methods, hearts from the two patients obtained at autopsy showed wider separation from non-failing donors than those that were obtained after transplantation (Fig. 1A–C). These findings highlight not only the clear distinction between non-failing donors and BTHS individuals but also the complexity and heterogeneity of BTHS, with various cellular processes being differentially affected in patients.

Next, we studied the top 25 metabolites, lipids and proteins sorted on *P*-value (Fig. 1D–F). Metabolite ranking is based on a comparison between non-failing donors and the fifteen BTHS samples (five individuals in biological triplicates) as a single group. This approach allows us to further confirm the robustness of our method by demonstrating high repeatability at the analyte level. For all subsequent statistical analyses, we use the average of each triplicate to avoid artificially inflating the sample size. Polar metabolites show homogeneity within patient replicates (Fig. 1D). However, their intensities are diverse between each individual, and several metabolites are significantly elevated in the samples collected at autopsy. In lipidomics, (monolyso-) cardiolipins dominate the list of significantly regulated lipids and show high uniformity in all BTHS samples (Fig. 1E). The PCA analysis demonstrates a high degree of separation even within the BTHS group, which hints at deeper effects on the lipidome not represented in this heatmap, as that shows a more uniform picture. Proteomics reveals a very clear distinction in protein abundances between controls and BTHS individuals, with the top significant changes being dominated by proteins from the NDUF family which belong to respiratory chain complex I (Fig. 1F). To illustrate the extent of changes within the metabolome, lipidome and proteome in BTHS heart tissue compared to non-failure control cardiac

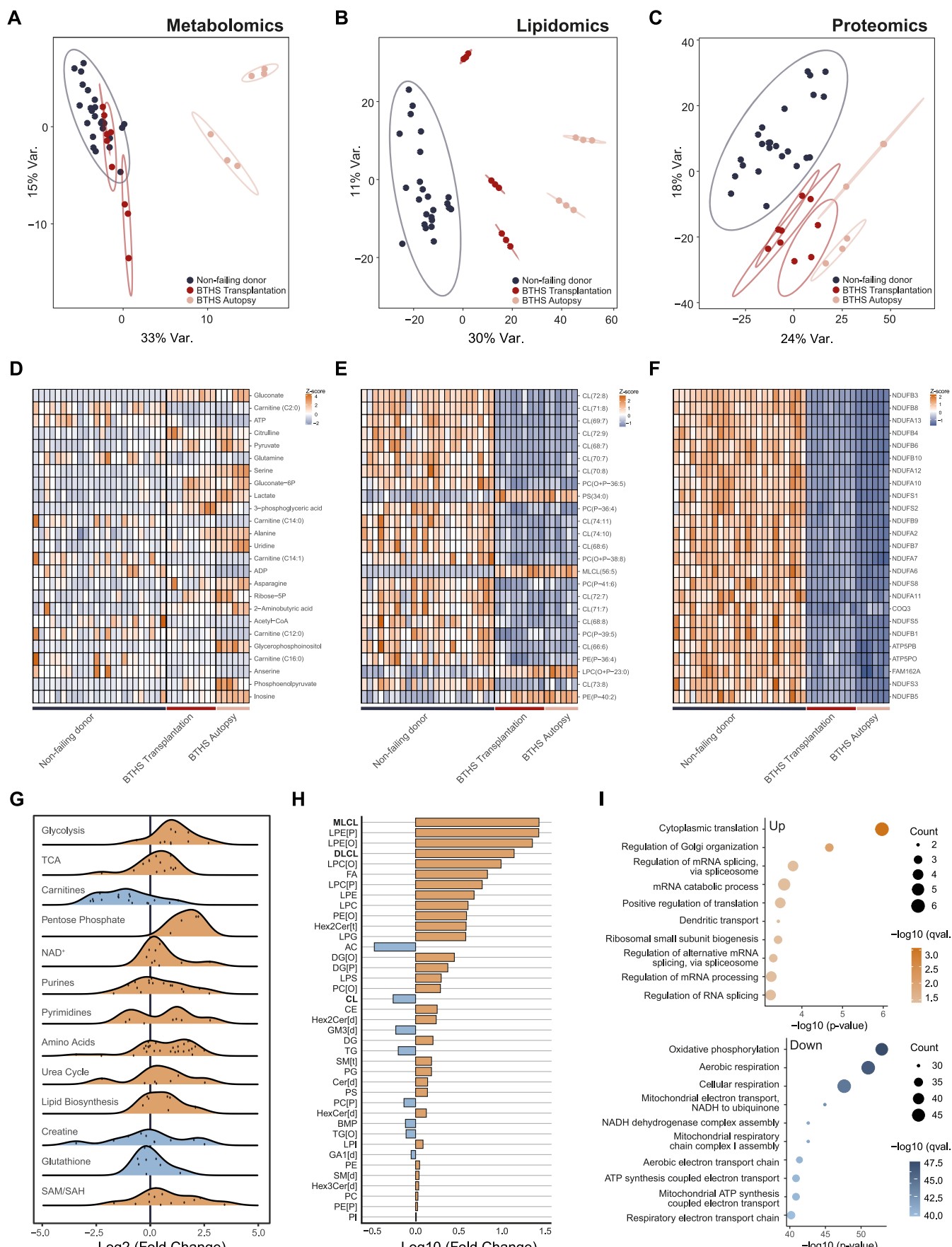

**Figure 1. Overview of phenotypic differences between BTHS ($n = 5$) and non-failing donors ($n = 24$) detected in metabolomics, lipidomics and proteomics.**

(A–C) Principal Component Analysis (PCA) for metabolomic (A), lipidomic (B) and proteomic (C) profiles, showing three technical BTHS replicates. Non-failing donors show distinct separation from all BTHS tissues, except for BTHS samples obtained during transplantation in metabolomics. BTHS tissues obtained at autopsy show a larger separation from non-failing donors than those obtained during transplantation in all analyses. (D–F) Heatmaps representing z-scores of the top 25 (ranked on P-value) metabolites (D), lipids (E) and proteins (F) in cardiac samples of BTHS individuals compared to non-failing donors. (G) Ridge plot of the changes in (polar) metabolic pathways shows pleiotropic differences between tissues of BTHS patients and non-failing donors, especially in energy-related pathways. (H) Lipid class totals across cardiac samples from BTHS individuals and non-failing donors, highlighting lipids related to the Tafazzin deficiency such as MLCL, DLCL and CL. (I) Bubble plot of enriched gene ontology (GO) biological processes for upregulated (top panel) and downregulated (bottom panel) proteins in BTHS cardiac tissue indicate that mitochondrial respiration is significantly decreased. Statistical comparisons were performed using moderated t-tests from the limma package in R, with Benjamini–Hochberg correction. Exact p-values corresponding to all statistical comparisons are provided in: Dataset EV4 (STATISTICS LIPIDOMICS), Dataset EV5 (STATISTICS METABOLOMICS), and Dataset EV6 (STATISTICS PROTEOMICS).

tissue, we compared both groups using metabolic pathways, lipid classes and GO-terms as classifying features, respectively (Fig. 1G–I). From the metabolomics data, differences are linked to pathways related to energy, such as glycolysis, short-chain carnitines and nucleotides (Fig. 1G), which corroborates the results in the top 25 altered metabolites. When surveying all lipids classes, CL and MLCL, directly related to the Tafazzin deficiency that causes BTHS are indeed highly affected (Fig. 1H). This overview reveals the depth of the alterations in the lipidome. Strikingly, many lysolipid-species of glycerophospholipids such as phosphatidylethanolamine (PE) and phosphatidylcholine (PC), including their alkyl and plasmalogen variants, are highly increased in abundance. While previous studies in the TAFAZZIN knockdown mouse heart showed unchanged PE[P] and reduced PC[P], our analysis in human heart tissue confirms these findings but additionally reveals elevated lyso-ether lipids (LPE[O], LPE[P], and LPC counterparts), which were not measured in the mouse model (Bozelli and Epand, 2022; Bozelli et al, 2020). In proteomics, GO-term enrichment analysis highlights the expected depletion in proteins related to mitochondrial respiration (Fig. 1I). Notably, there are far fewer proteins increased than decreased, and to a much lesser degree, with no clear trends. After determining that the replicates show only minor variance, their average abundances for metabolites, lipids or proteins were used for further specific analysis and statistics. Finally, to ensure that the observed differences in our study between non-failing donors and BTHS are disease-specific and not the result of potential confounders such as sex or age, we repeated the proteomics analysis that forms the core of most of our results, using only the five youngest male non-failing donors (Fig. EV1A). The top altered proteins show a highly similar signature as when compared to our total control group, highlighting that the observed differences are likely not affected by the varied differences of sex and age in our control group, but are true disease-specific differences in the BTHS heart.

## Confirmation of diagnostic markers CL, MLCL and DLCL in BTHS cardiac tissue

BTHS is caused by a deficiency in the cardiolipin remodelling enzyme Tafazzin (Fig. 2A). Clinical diagnosis of BTHS is therefore often initiated by determining the ratio of CL(72:8) and MLCL(50:2) in blood plasma or blood spots, as mature CL is depleted and the MLCL precursor is highly accumulated (Kulik et al, 2008). This distinctive profile is also clearly seen in cardiac tissue (Fig. 2B). A broader analysis reveals that the CL lipid cluster

shows major disruption (Fig. 2C), with a marked decrease in mature CL species, and a corresponding increase in immature CL species. Statistical analysis of individual MLCL and DLCL species is not possible, as these are below the limits of detection in healthy individuals. However, total MLCL and DLCL levels clearly illustrate their accumulation in BTHS affected individuals (Fig. 2D). Notably, for both the clinically validated biomarkers and the total levels of MLCL and DLCL, there are no obvious correlation between their levels and whether the heart sample was obtained at transplantation or post-mortem.

## Integrative omics reveals mitochondrial dysfunction in cardiac tissue from BTHS affected individuals

Mitochondrial dysfunction is a hallmark of BTHS and previous reports have shown structural abnormalities (Acehan et al, 2007; Xu et al, 2005) and OXPHOS dysfunction that result in energy depletion (Gonzalvez et al, 2013; Zegallai and Hatch, 2021) (Fig. 3A). As cardiolipin is a fundamental part of the mitochondrial membrane, we analysed lipid classes associated with the mitochondrial membrane from our lipidomics dataset (Basu Ball et al, 2018). Interestingly, only cardiolipin is strongly altered (Fig. 3B). As CL make up a significant part of the mitochondrial membrane, we analysed a subset of mitochondrial proteins according to the Mitocarta 3.0 database (Rath et al, 2021). This comparison highlights striking alterations in cardiac mitochondrial proteins from BTHS individuals (Fig. 3C). We repeated the analysis using only the youngest male non-failing donors ($n = 5$) to provide additional confidence that the observed changes are disease-specific (Fig. EV1B). The proteomic alterations remained highly consistent, underscoring the robustness of these changes in BTHS hearts. Cardiolipins play a critical role in stabilizing electron transport complexes (Houtkooper and Vaz, 2008) (Fig. 3A), and our GO-term analysis identified a significant reduction of OXPHOS proteins (Fig. 2I). Consistently, proteins associated with mitochondrial OXPHOS complexes are markedly decreased (Fig. 3D). In fact, a majority of proteins in complex I and V are reduced. Interestingly, all CoQ enzymes in the current dataset, involved in the production of CoQ10, an important electron carrier in the ETC, are significantly reduced (Fig. 3E). However, CoQ10 itself is maintained at normal levels. Finally, a reduction in energy carriers such as adenosine triphosphate (ATP) shows a clear trend (Fig. 3F). The same pattern is also observed in phosphocreatine, a phosphate storage for rapid ATP production, with two of the critical enzymes related to this phosphate exchange significantly decreased (Fig. 3G).

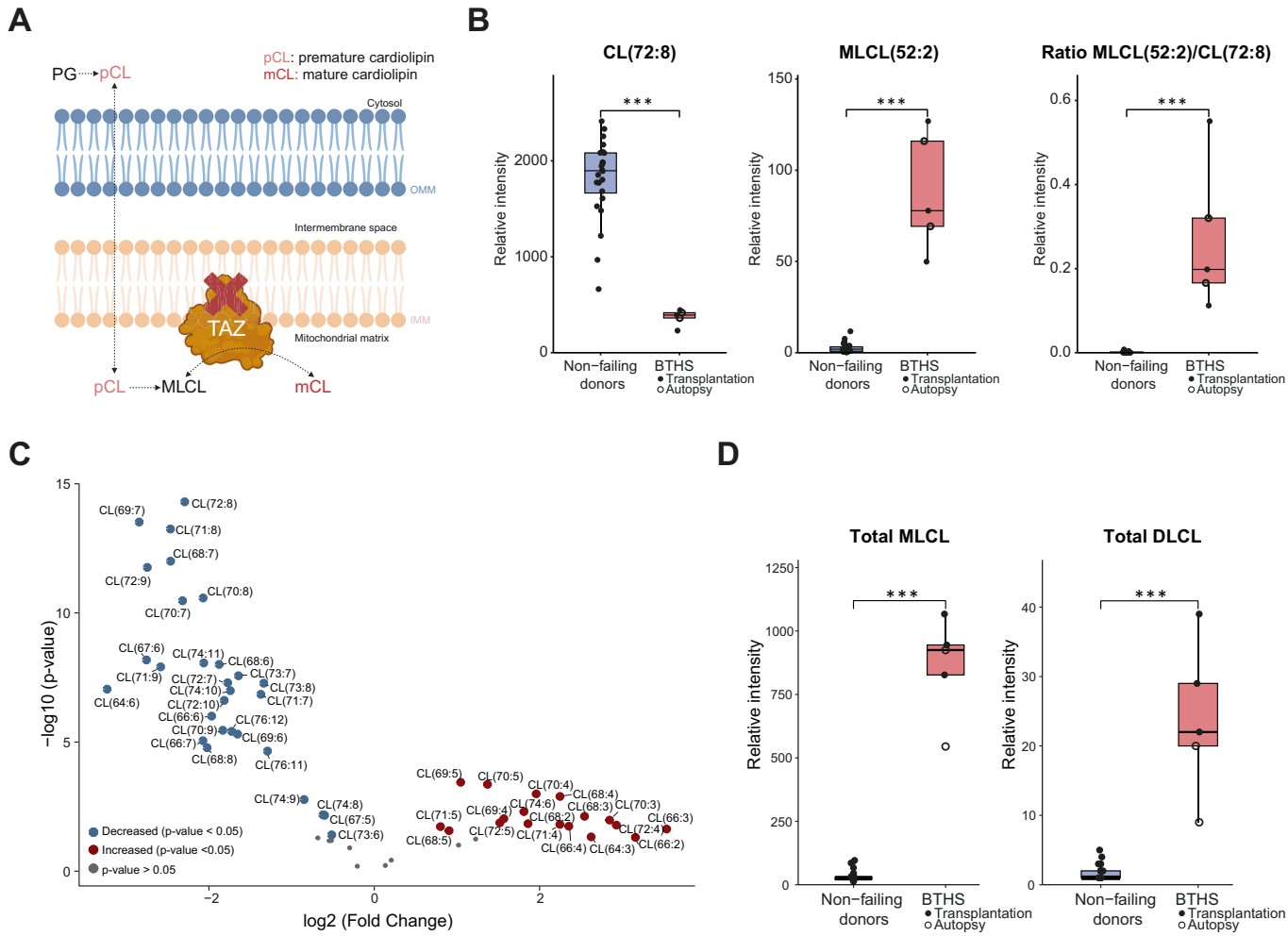

**Figure 2. Diagnostic markers CL, MLCL and DLCL in BTHS cardiac tissue.**

(A) Overview of cardiolipin biosynthesis and remodelling via the Tafazzin enzyme, which is dysfunctional in BTHS. (B) Relative intensity of CL(72:8), MLCL(50:2) and the ratio between MLCL(50:2) and CL(72:8) in both BTHS ($n = 5$) and non-failing donors ($n = 24$). Boxplots represent the interquartile range (IQR), with the box spanning the 25th to 75th percentiles and the median indicated by a horizontal line. Whiskers extend to the most extreme data points within 1.5×IQR. Statistical comparisons were performed using moderated t-tests from the limma package in R, with Benjamini–Hochberg correction. Annotated p-values are defined as follows: *$p < 0.05$, **$p < 0.01$, ***$p < 0.001$, ****$p < 0.0001$. (C) Volcano plot of CL species, depicting depleted CL species (blue) and accumulated CL species (red) in cardiac tissue of BTHS individuals ($n = 5$) compared to non-failing donors ($n = 24$). (D) Relative intensity of the total MLCL and DLCL classes in BTHS individuals ($n = 5$) and non-failing donors ($n = 24$). Boxplots are defined as in (B). Statistical comparisons were performed using moderated t-tests from the limma package in R, with Benjamini–Hochberg correction. Annotated p-values are defined as in (B). Exact p-values corresponding to all statistical comparisons are provided in: Dataset EV4 (STATISTICS LIPIDOMICS), Dataset EV5 (STATISTICS METABOLOMICS), and Dataset EV6 (STATISTICS PROTEOMICS).

## BTHS cardiac tissue shows a metabolic substrate shift

In a healthy heart, myocardial metabolism primarily relies on FAO to meet its high energy demand (Pascual and Coleman, 2016). As we observed severe depletion of mitochondrial proteins especially, as well as ATP and other energy-related metabolites, we studied the energy metabolism of BTHS tissues. Combining data from all three omics, we provide a comprehensive overview of cellular energy metabolism (Fig. 4), including glycolysis (Fig. 4A), the TCA cycle (Fig. 4B) and mitochondrial β-oxidation (Fig. 4C). We observed that acetyl-CoA, the main substrate of the TCA cycle, is significantly depleted in BTHS cardiac tissue. Acetyl-CoA can be produced through various pathways, with the predominant sources being FAO and glycolysis. In line with a metabolic shift away from

FAO, we observe a major and largely significant decrease of carnitines, independent of chain length and saturation. Several proteins related to mitochondrial β-oxidation show a decreased abundance. Very long-chain acyl-CoA dehydrogenase (VLCAD), responsible for the oxidation of long-chain fatty acids within mitochondria, showed the greatest reduction, followed by medium-chain acyl-CoA dehydrogenase (MCAD) whereas short-chain acyl-CoA dehydrogenase (SCAD) shows no changes. Strikingly, individuals with reduced levels of these proteins displayed an approximately two-fold increase in the trifunctional protein HADHA, a major component of mitochondrial β-xidation, whereas no such increase was observed in other BTHS individuals. Along with this apparent general decrease in FAO in BTHS, there is a complementary increase in the glycolytic end-products pyruvate

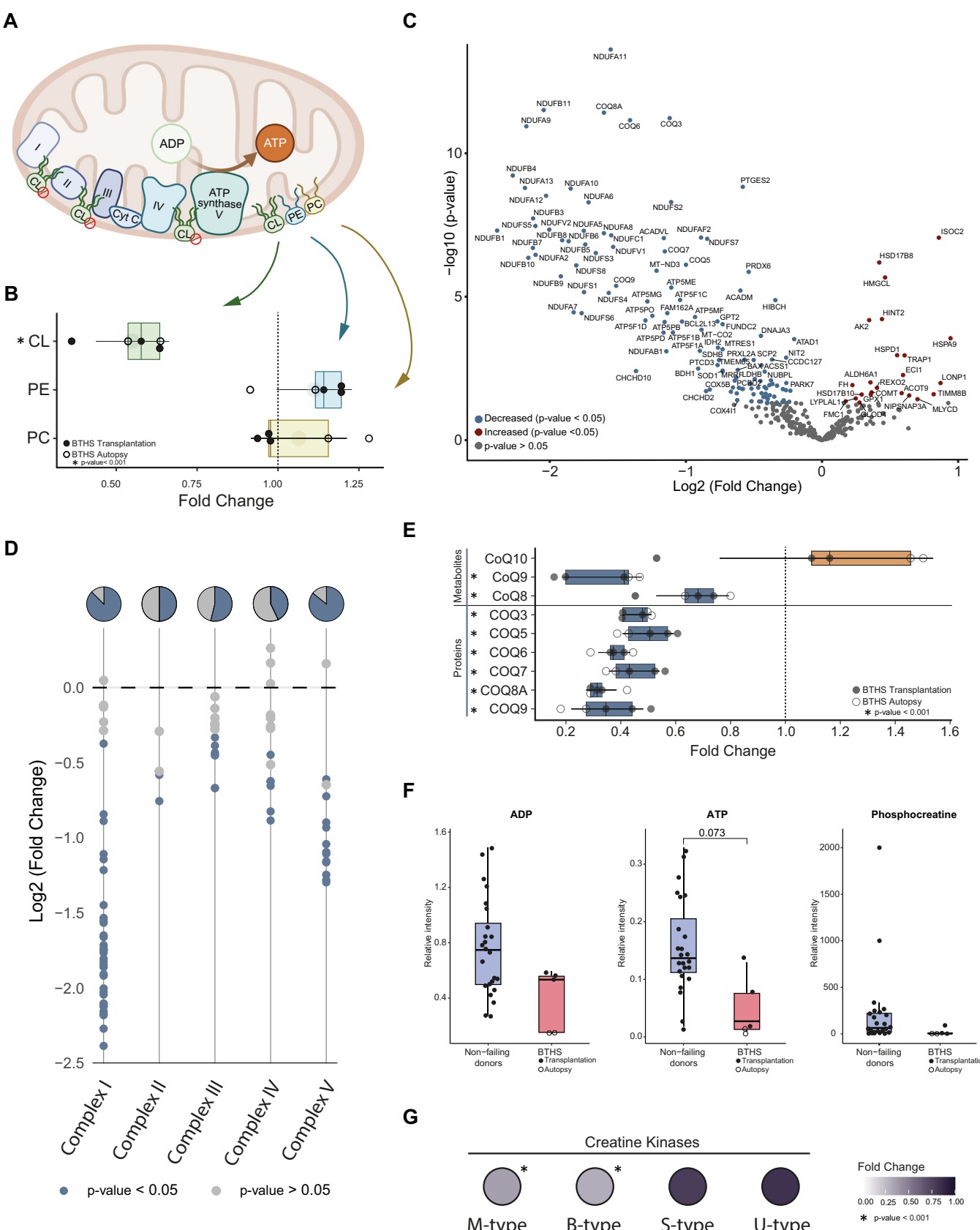

**Figure 3. Mitochondrial dysfunction in cardiac tissue from BTHS individuals.**

(A) Schematic overview of the mitochondria, highlighting the function of CL as a mitochondrial membrane lipid. In BTHS, CL depletion contributes to the destabilization of the ETC complexes which impairs ATP production via OXPHOS. (B) Changes in abundance of main mitochondrial membrane lipids in cardiac tissue of BTHS individuals ($n = 5$) compared to non-failing donors ($n = 24$). Boxplots represent the interquartile range (IQR), with the box spanning the 25th to 75th percentiles and the median indicated by a horizontal line. Whiskers extend to the most extreme data points within 1.5×IQR. Statistical comparisons were performed using moderated t-tests from the limma package in R, with Benjamini–Hochberg correction. (C) Volcano plot of mitochondrial proteins (according to MitoCarta 3.0), depicting depleted- (blue) and accumulated proteins (red) in cardiac tissue of BTHS individuals ($n = 5$) compared to non-failing donors ($n = 24$). (D) Changes in electron transport chain proteins show major dysfunction, and a majority of decreased proteins in Complex I and Complex V. The pie charts represent the proportion of decreased proteins in each complex. Statistical comparisons were performed using moderated t-tests from the limma package in R, with Benjamini–Hochberg correction. (E) Changes in coenzyme Q related metabolites and proteins, associated with the electron transport chain, in cardiac tissue of BTHS individuals ($n = 5$) compared to non-failing donors ($n = 24$). Boxplots are defined as in (B). Statistical comparisons were performed using moderated t-tests from the limma package in R, with Benjamini–Hochberg correction. (F) Relative intensity of ADP, ATP and phosphocreatine in both BTHS individuals ($n = 5$) and non-failing donors ($n = 24$). Boxplots are defined as in (B). Statistical comparisons were performed using moderated t-tests from the limma package in R, with Benjamini–Hochberg correction. (G) Changes in abundance of creatine kinase isoforms of M-type, B-type, S-type, U-type in BTHS cardiac tissues ($n = 5$) where colour intensity corresponds to the magnitude of change in comparison to non-failing donors ($n = 24$). Statistical comparisons were performed using moderated t-tests from the limma package in R, with Benjamini–Hochberg correction. Exact p-values corresponding to all statistical comparisons are provided in: Dataset EV4 (STATISTICS LIPIDOMICS), Dataset EV5 (STATISTICS METABOLOMICS), and Dataset EV6 (STATISTICS PROTEOMICS).

and lactate, as well as alanine. Interestingly, glycolysis enzyme abundance seems to show pleiotropic dysfunction. Hexokinase 1 and the pyruvate dehydrogenase complex show a trend towards upregulation, while four of the intermediate enzymes are significantly downregulated. Finally, TCA cycle metabolites appeared unaffected, while related enzymes show a trend of downregulation. Although considerable inter-individual variation was observed, the results suggest an altered substrate preference from FAO to glycolysis in BTHS heart tissue.

## Cardiac remodelling in BTHS hearts

BTHS patients often present with diminished cardiac performance, though individual phenotypes are diverse (Taylor et al, 2022; Dudek and Maack, 2017). To investigate whether BTHS hearts undergo structural remodelling, we analysed actin-myosin cytoskeleton proteins (Fig. 5A) and compared their abundance between BTHS tissues obtained at autopsy and during transplantation (Fig. 5B). BTHS hearts exhibit numerous altered proteins associated with cardiac structure as well as electrical and mechanical function. As this analysis can be sensitive to confounders due to the low sample numbers and inherent sampling differences, we also investigated if the failing hearts showed a more generic apoptotic/necrotic phenotype using a subset of established protein markers. No distinct pattern was observed (Fig. EV2A). However, metabolic changes associated with remodelling and heart failure were observed in BTHS hearts. Alterations in branched-chain amino acid (BCAA) metabolism have been observed in HF models (Lai et al, 2014; Sansbury et al, 2014). BCAA were markedly increased in the two deceased individuals, compared to both controls and samples obtained at transplantation (Fig. 5C). Enzymes related to the metabolism of BCAAs also exhibit a noticeable difference between deceased and other individuals (Fig. 5D). Furthermore, lipidomics revealed substantial accumulation of heart failure associated lipids, mostly in deceased individuals, such as sphingosines (Fig. 5E), ceramides (Fig. 5F) and fatty acids (Fig. 5G). These types of lipids can contribute to lipotoxicity when accumulated in cardiac tissue (Da Dalt et al, 2023; Schulze et al, 2016; Stratford et al, 2004; Borradaile et al, 2006). In BTHS, total sphingosine and ceramide levels were doubled in individuals with failing hearts compared to non-failing controls. Interestingly, these levels were within the normal range for BTHS

patients without heart failure. Fatty acids were increased in all BTHS tissues, but were significantly increased in deceased individuals. When looking at the composition of these lipids, mostly the shorter sphingosines appear affected, while ceramides and fatty acids are more uniformly increased.

## Discussion

The aim of this study was to analyse the metabolome, lipidome and proteome of BTHS cardiac tissue and gain insights into the pathological mechanisms behind BTHS cardiomyopathy in human hearts. Our combined triple-omics platform is a powerful analytical tool since the direct comparison and integration of the multi-omics results is greatly aided by the fact that all omics analyses were performed on a single sample of heart tissue. This approach allowed us to achieve a comprehensive molecular and metabolic characterization of BTHS cardiac tissue with a high degree of certainty, illustrated by the robustness of our BTHS triplicate analysis. Notably, within the BTHS patient group, inter-individual differences detected by our metabolomics analysis are larger than those found in our lipidomics or proteomics. This is consistent with our expectations. The primary defect in BTHS lies within the lipid metabolism (specifically, cardiolipin biosynthesis) which leads to a relatively uniform and pronounced alteration in the lipidome across patients. Given cardiolipins central role in maintaining mitochondrial membrane structure, it is not surprising that proteins dependent on membrane integrity, such as those in the electron transport chain, are consistently affected, as reflected in our proteomics data and previous research (Huang et al, 2015). In contrast, the polar metabolome is further downstream from the primary defect and is influenced by a broader range of factors, including individual compensatory mechanisms and natural metabolic variability. This results in a more heterogeneous metabolic profile among BTHS patients. Nevertheless, we still observe a distinct metabolic signature for BTHS patients when compared to non-failing donors, for instance in pathways related to energy metabolism, which is consistent with other studies, clinical observations, and our other data. Corroborating earlier findings, our method detected the effects of the primary defect, accumulation of MLCL (and DLCL), CL depletion (Houtkooper et al, 2009a, 2009b).

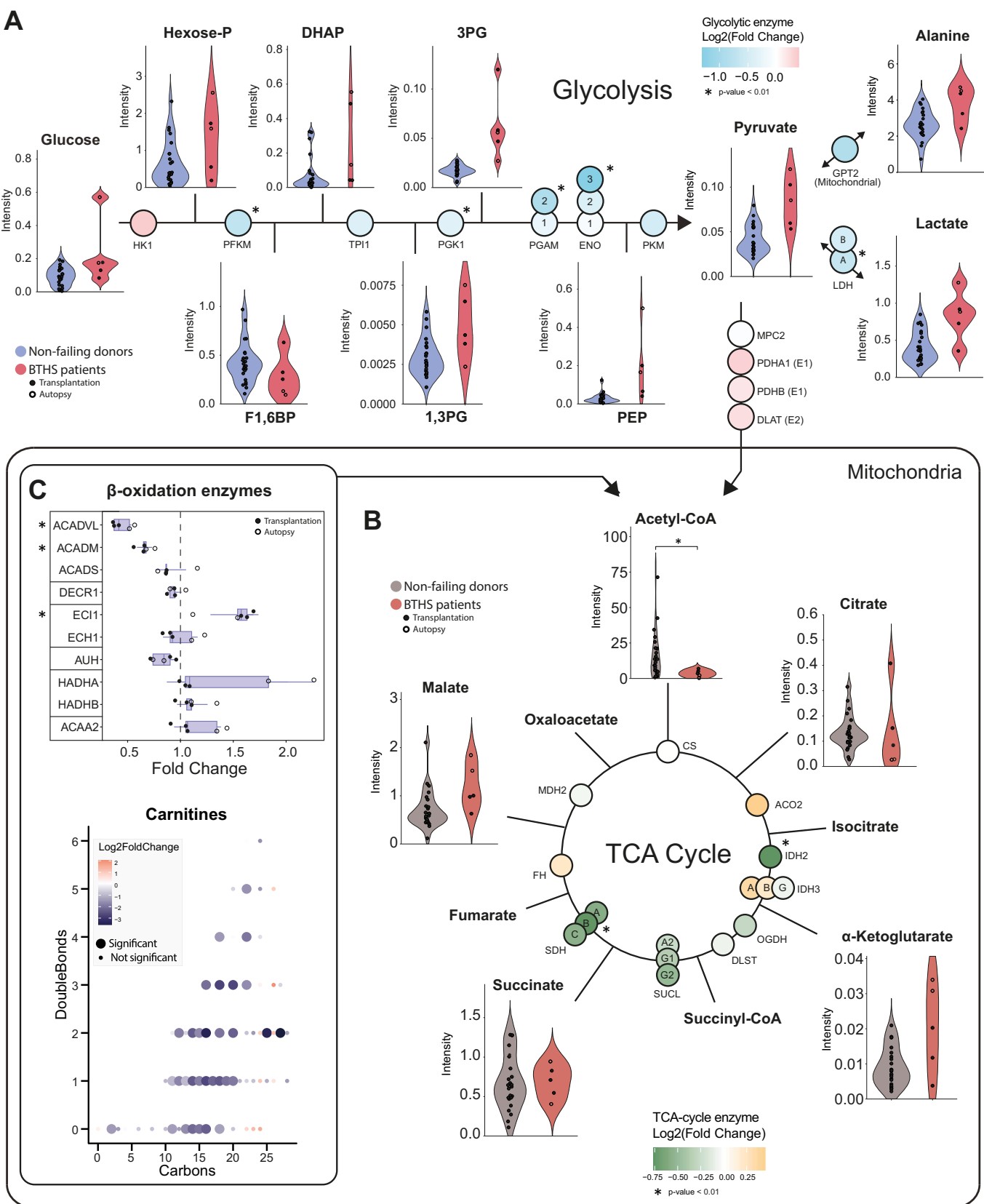

**Figure 4. Comprehensive overview of energy metabolism in BTHS cardiac tissue, integrating metabolomics, lipidomics and proteomics data.**

The interconnections between (**A**) glycolysis, (**B**) the TCA cycle, and (**C**) β-oxidation are shown. For glycolysis and the TCA cycle, relative metabolite intensity is represented in the violin plots comparing the non-failing donors ($n = 24$) and the BTHS group ($n = 5$). Changes in the abundance of enzymes are represented using a colour gradient in the circles between metabolites. β-Oxidation enzymes are shown as horizontal boxplots, and an overview is provided of changes in carnitine species. Boxplots represent the interquartile range (IQR), with the box spanning the 25th to 75th percentiles and the median indicated by a horizontal line. Whiskers extend to the most extreme data points within 1.5×IQR. Statistical comparisons were performed using moderated t-tests from the limma package in R, with Benjamini–Hochberg correction. Annotated $p$-values are defined as follows: $*p < 0.05$, $**p < 0.01$, $***p < 0.001$, $****p < 0.0001$. Exact $p$-values corresponding to all statistical comparisons are provided in: Dataset EV4 (STATISTICS LIPIDOMICS), Dataset EV5 (STATISTICS METABOLOMICS), and Dataset EV6 (STATISTICS PROTEOMICS). F1,6BP Fructose-1,6-biphosphate; DHAP Dihydroxyacetone phosphate; 1,3PG 1,3-phosphoglycerate; 3PG 3-phosphoglycerate; PEP Phosphoenolpyruvate.

Observed across all three datasets, BTHS heart samples showed clear signs of mitochondrial dysfunction and energy depletion. The proteome of BTHS heart samples showed the NDUF protein family significantly decreased. These proteins are subunits of respiratory complex I, and their depletion is a known hallmark in BTHS, previously reported in lymphoblasts (McKenzie et al, 2006) and fibroblasts (Chatzispyrou et al, 2018) from BTHS individuals. Additionally, proteins in the complexes I to V were severely decreased. This was expected since the absence of mature CL, or accumulation of MLCL, can impact mitochondrial inner membrane architecture and also the integrity and enzymatic activity of the ETC supercomplexes that generate ATP through OXPHOS (Huang et al, 2015; Dudek et al, 2013, 2016). Changes in the CL pool have also been causally linked to structural abnormalities of the ETC supercomplexes in heart failure (Sparagna et al, 2007; Saini-Chohan et al, 2009). In the metabolic profile, we also identified the downstream effects with ADP, ATP and creatine phosphate showing noticeable, though not significant, decreases in BTHS. The latter is known to be an energy buffer to ensure myofilament contractility in heart tissue and the autopsy samples from BTHS individuals seem to be most affected by this apparent inability to maintain energy homeostasis (Dudek and Maack, 2022).

Considering the significant metabolic flexibility the human heart has to properly adapt to substrate availability and uptake, we investigated sugar and fat metabolism (Da Dalt et al, 2023; Kolwicz et al, 2013). The heart relies mainly on fatty acids for ATP production (60–90%) (Pascual and Coleman, 2016), and a shift from FAO towards glucose oxidation can represent a maladaptive compensatory mechanism (Bertero and Maack, 2018). In this study, BTHS cardiac tissues showed profound and pleiotropic effects associated with mitochondrial function and energy production. We observed a reduction in FAO-related metabolites and enzymes in BTHS heart samples. This mechanism was described earlier in a BTHS mouse model, identified in cardiomyocytes derived from BTHS patient-induced pluripotent stem cells (iPSC) (Chowdhury et al, 2023) and suggested to be directly associated with worse cardiac function in young adults with BTHS (Cade et al, 2021).

Metabolic remodelling and structural remodelling are tightly connected in heart tissue (Bertero and Maack, 2018), which was also observed in this study. The actin-myosin cytoskeleton showed signs of structural remodelling when comparing heart tissue samples obtained at autopsy to those obtained at transplantation. These alterations were in line with findings from BTHS patient-derived cellular models, where mitochondrial abnormalities have been reported to cause sarcomere disarray and subsequent defective contractility (Wang et al, 2014).

We observed alterations in amino acid metabolism in the BTHS heart samples. However, these abnormalities were not consistent for all BTHS samples, as autopsy-derived and transplantation-derived samples clustered into two distinct groups. Notably, irregular myocardial amino acid metabolism has been reported in non-BTHS models of heart failure (Lai et al, 2014; Sansbury et al, 2014). Furthermore, BCAA have been reported to accumulate in human DCM heart tissue (Uddin et al, 2019). Consistent with this, our analysis revealed that BCAA levels were most significantly altered in tissues from deceased patients. This trend was mirrored by similar alterations in enzymes levels involved in BCAA metabolism.

A variety of metabolites and lipids associated with heart remodelling and failure were found to accumulate in BTHS heart samples, corroborating earlier findings in heart failure studies (Huang et al, 2022). Sphingolipids and ceramides have been linked to cardiac lipotoxicity as they alter cellular membrane organization, inducing cardiomyocyte apoptosis (Da Dalt et al, 2023; Stratford et al, 2004). Accumulated fatty acids contribute to lipotoxicity increasing ROS production that causes endoplasmic reticulum (ER) stress (Schulze et al, 2016; Borradaile et al, 2006). However, our integrated omics data also highlights the heterogeneity within the BTHS group. This accumulation of toxic lipids was especially clear in BTHS hearts collected at autopsy when compared to BTHS heart samples collected at transplantation, providing a possible lead for biomarkers related to disease severity.

A limitation of this study is the unavailability of age- and sex-matched heart tissue donor material. Due to the exceptionally young age at which BTHS heart failure occurs, matching cardiac material from healthy donors is exceedingly rare. Thus, our control group consists of heart tissue from a wide variety of phenotypes (male/females and three different age groups). Despite this variability, we still identified consistent differences between both groups, highlighting a distinct BTHS phenotype. However, due to the diversity of our control group, more general markers of cardiac and mitochondrial dysfunction might not be identified to their full extent in the current comparisons. For instance, this could support the observed lack of glutathione depletion or clear shift towards glycolysis in BTHS hearts when compared to non-failing donors, despite severe mitochondrial dysfunction and likely lipotoxicity in BTHS hearts.

Clinical heterogeneity is a known feature inherent in BTHS (Taylor et al, 2022) and the present study includes a relatively small sample size due to the inherent rarity of BTHS heart tissue samples. Two of these samples were obtained at autopsy and three at transplantation, thus likely under different circumstances. Unfortunately, patient information is limited and information such as *TAFAZZIN* genotype, medication and cardiac phenotype at the time of collection are unknown (Chatfield et al, 2022).

Despite these limitations, our approach provides an unprecedented amount of high-quality data on the BTHS cardiac tissue, encompassing broad swaths of the heart tissues' biological organization. The findings are consistent with prior studies on BTHS model systems and collectively reveal the extensive metabolic, lipidomic and proteomic alterations in BTHS heart tissues, serving as a robust foundation for future explorations into targeted treatment development.

In conclusion, we analysed the myocardial metabolic, lipidomic and proteomic profile of a single cardiac biopsy of individuals affected with BTHS using a unified extraction providing high-quality data, confirming known BTHS aberrations in heart tissue, while also providing potentially new areas of interest.

# Methods

### Reagents and tools table

| Reagent/Resource | Reference or Source | Identifier or Catalog number |
|---|---|---|
| **Experimental models** | | |
| Non-Failing donor cardiac tissue | Department of Physiology and Cell Biology, College of Medicine, The Ohio State University, Columbus, Ohio, USA | NA |
| Barth syndrome cardiac tissue | Barth Syndrome Foundation, 2005 Palmer Avenue #1033 Larchmont, NY 10538 | NA |
| **Recombinant DNA** | | |
| **Antibodies** | | |
| **Oligonucleotides and other sequence-based reagents** | | |
| **Chemicals, Enzymes and other reagents** | | |
| Lactic acid-D$_3$ | Millipore-Sigma (sigmaaldrich.com) | Cat#616702 |
| $^{13}$C$_3$-Pyruvate | Millipore-Sigma (sigmaaldrich.com) | Cat#490717 |
| $^{13}$C$_6$-D-Glucose | Millipore-Sigma (sigmaaldrich.com) | Cat#389374 |
| $^{13}$C$_6$-D-Glucose--6-phosphate | Cambridge Isotope Laboratories, Inc. (isotope.com) | Cat#CLM-8367 |
| $^{13}$C$_6$-D-Fructose-1,6-diphosphate | Cambridge Isotope Laboratories, Inc. (isotope.com) | Cat#CLM-8962 |
| Adenosine-$^{15}$N$_5$-triphosphate | Millipore-Sigma (sigmaaldrich.com) | Cat#707783 |
| Guanosine-$^{15}$N$_5$-triphosphate | Millipore-Sigma (sigmaaldrich.com) | Cat#707775 |
| Adenosine-$^{15}$N$_5$-monophosphate | Millipore-Sigma (sigmaaldrich.com) | Cat#662658 |
| Guanosine-$^{15}$N$_5$-monophosphate | Millipore-Sigma (sigmaaldrich.com) | Cat#662674 |
| Succinic acid-D$_6$ | Millipore-Sigma (sigmaaldrich.com) | Cat#488356 |
| Citric acid-D$_4$ | Millipore-Sigma (sigmaaldrich.com) | Cat#485438 |

| Reagent/Resource | Reference or Source | Identifier or Catalog number |
|---|---|---|
| L-Carnitine-D$_3$ | Millipore-Sigma (sigmaaldrich.com) | Cat#616737 |
| Thymine-D$_4$ | Millipore-Sigma (sigmaaldrich.com) | Cat#487066 |
| $^{13}$C$_2$-Glycine | Millipore-Sigma (sigmaaldrich.com) | Cat#283827 |
| Alanine-D$_4$ | Millipore-Sigma (sigmaaldrich.com) | Cat#485845 |
| Arginine-D$_7$ | Millipore-Sigma (sigmaaldrich.com) | Cat#776408 |
| Aspartate-D$_3$ | Millipore-Sigma (sigmaaldrich.com) | Cat#589667 |
| $^{13}$C$_1$-Citrulline | Cambridge Isotope Laboratories, Inc. (isotope.com) | Cat#CLM-4899 |
| Glutamate-D$_3$ | Millipore-Sigma (sigmaaldrich.com) | Cat#749435 |
| Glutamine-D$_5$ | Millipore-Sigma (sigmaaldrich.com) | Cat#616303 |
| $^{13}$C$_6$-Isoleucine | Millipore-Sigma (sigmaaldrich.com) | Cat#604798 |
| Leucine-D$_3$ | Millipore-Sigma (sigmaaldrich.com) | Cat#486825 |
| Lysine-D$_4$ | Millipore-Sigma (sigmaaldrich.com) | Cat#489034 |
| Methionine-D$_3$ | Millipore-Sigma (sigmaaldrich.com) | Cat#300616 |
| Ornithine-D$_6$ | Millipore-Sigma (sigmaaldrich.com) | Cat#749443 |
| Phenylalanine-D$_5$ | Cambridge Isotope Laboratories, Inc. (isotope.com) | Cat#DLM-1258 |
| Proline-D$_7$ | Cambridge Isotope Laboratories, Inc. (isotope.com) | Cat#DLM-487 |
| Serine-D$_3$ | Millipore-Sigma (sigmaaldrich.com) | Cat#688436 |
| Tryptophan-D$_5$ | Millipore-Sigma (sigmaaldrich.com) | Cat#615862 |
| Tyrosine-D$_4$ | Millipore-Sigma (sigmaaldrich.com) | Cat#792721 |
| Valine-D$_8$ | Millipore-Sigma (sigmaaldrich.com) | Cat#486612 |
| EasyPep™ MS Sample Prep Kits | ThermoFisher Scientific | A45733 |
| Bis(monoacylglycero)phosphate BMP(14:0)2 | Avanti Polar Lipids, Alabaster, AL, USA | 857131 |
| D7-Cholesteryl Ester CE(16:0) | Avanti Polar Lipids, Alabaster, AL, USA | 700149 |
| Cardiolipin CL(14:0)4 | Avanti Polar Lipids, Alabaster, AL, USA | 750332 |
| Diacylglycerol DAG(14:0)2 | Avanti Polar Lipids, Alabaster, AL, USA | 800814 |
| Glucose Ceramide GlcCer(d18:1/12:0) | Avanti Polar Lipids, Alabaster, AL, USA | 860543 |

| Reagent/Resource | Reference or Source | Identifier or Catalog number |
|---|---|---|
| Lysophosphatidic acid LPA(14:0) | Avanti Polar Lipids, Alabaster, AL, USA | 857120 |
| Lysophosphatidylcholine LPC(14:0) | Avanti Polar Lipids, Alabaster, AL, USA | 855575 |
| Lysophosphatidylethanolamine LPE(14:0) | Avanti Polar Lipids, Alabaster, AL, USA | 856735 |
| Lysophosphatidylglycerol LPG(14:0) | Avanti Polar Lipids, Alabaster, AL, USA | 858120 |
| Phosphatidic acid PA(14:0)2 | Avanti Polar Lipids, Alabaster, AL, USA | 830845 |
| Phosphatidylcholine PC(14:0)2 | Avanti Polar Lipids, Alabaster, AL, USA | 850345 |
| Phosphatidylethanolamine PE(14:0)2 | Avanti Polar Lipids, Alabaster, AL, USA | 850745 |
| Phosphatidylglycerol PG(14:0)2 | Avanti Polar Lipids, Alabaster, AL, USA | 840445 |
| Phosphatidylinositol PI(8:0)2 | Avanti Polar Lipids, Alabaster, AL, USA | 850181 |
| Phosphatidylserine PS(14:0)2 | Avanti Polar Lipids, Alabaster, AL, USA | 840033 |
| Sphingomix (includes Cer(d18:1/12:0), Cer(d18:1/25:0), SM(d18:1/12:0), SPH(d17:0), SPH(d17:1), S1P(d17:0), S1P(d17:1)) | Avanti Polar Lipids, Alabaster, AL, USA | LM600 |
| **Other** | | |
| Waters Acquity UPLC | Waters (waters.com) | N/A |
| Ultimate 3000 UPLC | ThermoFisher Scientific | N/A |
| Ultra-high resolution LC-QTOF MS Impact II | Bruker (bruker.com) | N/A |
| TASQ® Software (version 2025.1.1 7c91baf6) | Bruker (bruker.com) | N/A |
| DIA-NN | https://github.com/vdemichev/DiaNN | 1.8.1 |
| timsTOF Pro 2 | Bruker (bruker.com) | N/A |
| SeQuant ZIC-cHILIC column (PEEK 100 × 2.1 mm, 3 μm) | Merck Millipore | 150657 |
| LUNA silica column (250 × 2 mm, 5 μm 100 Å) | Phenomenex | 00G-4274-B0 |
| HSS T3 column (150 × 2.1 mm, 1.8 μm) | Waters | 186003540 |
| Acquity UPLC BEH C18 Column (130 Å, 1.7 μm, 2.1 mm × 50 mm) | Waters | 186002350 |
| Acquity UPLC Vanguard BEH C18 precolumn | Waters | 186003975 |

## Patient samples

Five left ventricular heart samples from individuals affected by BTHS cardiomyopathy were used in this study (Chatfield et al, 2022). In Appendix Table S1, the information about each sample is outlined. The samples were provided by the Barth Syndrome Foundation DNA and Tissue Bank. The samples from BTHS individuals were compared to a group of samples from non-failing donors (female/male; 18–>55 years; Caucasian/African-American) provided by Ohio State University. BTHS triplicates, prepared as separate samples from the same donor tissue, were created to allow for the assessment of both biological and technical variance.

The study protocol was reviewed by Medical Ethics Review Committee of the Academic Medical Center (Amsterdam UMC) and deemed not subject to the Dutch Medical Research Involving Human Subjects Act (WMO), hence formal committee approval was not required (letter dated March 25, 2021, reference W21_148 # 21.163). Informed consent was obtained from all subjects included in the study and experiments were conformed to the principles set out in the World Medical Association (WMA) Declaration of Helsinki and the Department of Health and Human Services Belmont Report. Barth syndrome samples were obtained from University of Florida, Barth Syndrome Registry and DNA Bank (IRB201702372). The control samples were obtained in collaboration with The Lifeline of Ohio and The Ohio State University under established and approved protocols.

## Multi-omics sample preparation

Metabolomics, lipidomics and proteomics were performed after a unified extraction method, combining several established methods (Nakayasu et al, 2016; Molenaars et al, 2021; Szyrwiel et al, 2024; Schomakers et al, 2022). In a 2 mL tube, containing approximately 3 mg of freeze-dried heart tissue, the following amounts of internal standard dissolved in water were added to each sample for metabolomics: adenosine-$^{15}N_5$-monophosphate (5 nmol), adenosine-$^{15}N_5$-triphosphate (5 nmol), $D_4$-alanine (0.5 nmol), $D_7$-arginine (0.5 nmol), $D_3$-aspartic acid (0.5 nmol), $D_3$-carnitine (0.5 nmol), $D_4$-citric acid (0.5 nmol), $^{13}C_1$-citrulline (0.5 nmol), $^{13}C_6$-fructose-1,6-diphosphate (1 nmol), $^{13}C_2$-glycine (5 nmol), guanosine-$^{15}N_5$-monophosphate (5 nmol), guanosine-$^{15}N_5$-triphosphate (5 nmol), $^{13}C_6$-glucose (10 nmol), $^{13}C_6$-glucose-6-phosphate (1 nmol), $D_3$-glutamic acid (0.5 nmol), $D_5$-glutamine (0.5 nmol), $^{13}C_6$-isoleucine (0.5 nmol), $D_3$-lactic acid (1 nmol), $D_3$-leucine (0.5 nmol), $D_4$-lysine (0.5 nmol), $D_3$-methionine (0.5 nmol), $D_6$-ornithine (0.5 nmol), $D_5$-phenylalanine (0.5 nmol), $D_7$-proline (0.5 nmol), $^{13}C_3$-pyruvate (0.5 nmol), $D_3$-serine (0.5 nmol), $D_6$-succinic acid (0.5 nmol), $D_4$-thymine (1 nmol), $D_5$-tryptophan (0.5 nmol), $D_4$-tyrosine (0.5 nmol), $D_8$-valine (0.5 nmol).

In the same 2 mL tube, the following amounts of internal standards dissolved in 1:1 (v/v) methanol:chloroform were added for lipidomics: Bis(monoacylglycero)phosphate BMP(14:0)2 (0.2 nmol), Ceramide-1-phosphate C1P(d18:1/12:0) (0.125 nmol), $D_7$-Cholesteryl Ester CE(16:0) (2.5 nmol), Ceramide Cer(d18:1/12:0) (0.125 nmol), Ceramide Cer(d18:1/25:0) (0.125 nmol), Cardiolipin CL(14:0)4 (0.1 nmol), Diacylglycerol DAG(14:0)2 (0.5 nmol), Glucose Ceramide GlcCer(d18:1/12:0) (0.125 nmol), Lactose Ceramide LacCer(d18:1/12:0) (0.125 nmol), Lysophosphatidicacid LPA(14:0) (0.1 nmol), Lysophosphatidylcholine LPC(14:0) (0.5 nmol), Lysophosphatidylethanolamine LPE(14:0) (0.1 nmol), Lysophosphatidylglycerol LPG(14:0) (0.02 nmol), Phosphatidic acid PA(14:0)$_2$ (0.5 nmol), Phosphatidylcholine PC(14:0)$_2$ (2 nmol), Phosphatidylethanolamine PE(14:0)$_2$ (0.5 nmol), Phosphatidylglycerol PG(14:0)$_2$ (0.1 nmol), Phosphatidylinositol PI(8:0)$_2$ (0.5 nmol), Phosphatidylserine PS(14:0)$_2$ (5 nmol), Sphinganine

1-phosphate S1P(d17:0) (0.125 nmol), Sphinganine-1-phosphate S1P(d17:1) (0.125 nmol), Ceramide phosphocholines SM(d18:1/12:0) (2.125 nmol), Sphingosine SPH(d17:0) (0.125 nmol), Sphingosine SPH(d17:1) (0.125 nmol), Triacylglycerol TAG(14:0)$_3$ (0.5 nmol). Subsequently, solvents were added to achieve a total volume of 500 μL water, 500 μL methanol. Muscle tissues were homogenized using a Qiagen TissueLyser II for 2 min at 30 times/s with a 5 mm Qiagen Stainless Steel Bead in each tube. Chloroform was added to each tube for a total chloroform volume of 1 mL and samples were thoroughly mixed, before centrifugation for 10 min at 14,000 rpm to facilitate layer separation.

## Metabolomics

The top layer, containing the polar phase, was transferred to a new 1.5 mL tube and dried using a vacuum concentrator at 60 °C. Dried samples were reconstituted in 100 μL 6:4 (v/v) methanol:water. Metabolites were analysed using a Waters Acquity ultra-high performance liquid chromatography system coupled to a Bruker Impact II™ Ultra-High Resolution Qq-Time-Of-Flight mass spectrometer. Samples were kept at 12 °C during analysis and 5 μL of each sample was injected. Chromatographic separation was achieved using a Merck Millipore SeQuant ZIC-cHILIC column (PEEK 100 × 2.1 mm, 3 μm particle size). Column temperature was held at 30 °C. Mobile phase consisted of (A) 1:9 (v/v) acetonitrile:water and (B) 9:1 (v/v) acetonitrile:water, both containing 5 mmol/L ammonium acetate. Using a flow rate of 0.25 mL/min, the LC gradient consisted of: Dwell at 100% Solvent B, 0–2 min; Ramp to 54% Solvent B at 13.5 min; Ramp to 0% Solvent B at 13.51 min; Dwell at 0% Solvent B, 13.51–19 min; Ramp to 100% B at 19.01 min; Dwell at 100% Solvent B, 19.01–19.5 min. Column was equilibrated by increasing flow rate to 0.4 mL/min at 100% B for 19.5–21 min. MS data were acquired using negative and positive ionization in full scan mode over the range of m/z 50–1200. Data were analysed using Bruker TASQ software version 2.1.22.3. All reported metabolite intensities were normalized to freeze-dried tissue weight, as well as to internal standards with comparable retention times and response in the MS. Metabolite identification was based on a combination of accurate mass, (relative) retention times and fragmentation spectra, compared with the analysis of a library of standards.

## Lipidomics

The bottom layer, containing the apolar phase, was transferred to a new 1.5 mL tube and evaporated under nitrogen at 60 °C. The residue was dissolved in 100 μL of 1:1 (v/v) methanol:chloroform. Lipids were analysed using a Thermo Scientific Ultimate 3000 binary HPLC coupled to a Q Exactive Plus Orbitrap mass spectrometer. For normal phase separation, 2 μL of each sample was injected onto a Phenomenex® LUNA silica, 250 * 2 mm, 5 μm 100 Å. Column temperature was held at 25 °C. Mobile phase consisted of (A) 85:15 (v/v) methanol:water containing 0.0125% formic acid and 3.35 mmol/L ammonia and (B) 97:3 (v/v) chloroform:methanol containing 0.0125% formic acid. Using a flow rate of 0.3 mL/min, the LC gradient consisted of: Dwell at 10% A 0–1 min, ramp to 20% A at 4 min, ramp to 85% A at 12 min, ramp to 100% A at 12.1 min, dwell at 100% A 12.1–14 min, ramp to 10% A at 14.1 min, dwell at 10% A for 14.1–15 min. For reversed

phase separation, 5 μL of each sample was injected onto a Waters HSS T3 column (150 × 2.1 mm, 1.8 μm particle size). Column temperature was held at 60 °C. Mobile phase consisted of (A) 4:6 (v/v) methanol:water and B 1:9 (v/v) methanol:isopropanol, both containing 0.1% formic acid and 10 mmol/L ammonia. Using a flow rate of 0.4 mL/min, the LC gradient consisted of: Dwell at 100% A at 0 min, ramp to 80% A at 1 min, ramp to 0% A at 16 min, dwell at 0% A for 16–20 min, ramp to 100% A at 20.1 min, dwell at 100% A for 20.1–21 min. MS data were acquired using negative and positive ionization using continuous scanning over the range of m/z 150 to m/z 2000. Data were analysed using an in-house developed lipidomics pipeline written in the R programming language (http://ww.r-project.org), as previously described (Jaspers et al, 2024). All reported lipids were normalized to corresponding internal standards according to lipid class, as well as to freeze-dried tissue weight. Lipid identification has been based on a combination of accurate mass, (relative) retention times, fragmentation spectra, analysis of samples with known metabolic defects, and the injection of relevant standards.

## Proteomics

After the transfer of both solvent layers, the remaining protein pellet was dried under a stream of nitrogen. Proteomics sample preparation was performed using a Thermo Scientific™ EasyPep™ MS Sample Prep Kit (A40006), according to the kit's instructions. Briefly, 200 μL lysis buffer was added to each sample and a Thermo Scientific™ Pierce™ BCA Protein Assay (23225) was performed according to kit instructions to determine protein content. For each sample, 100 μg of protein was transferred to a new 2 mL tube. Samples were reduced and alkylated at 95 °C for 10 min, followed by a two-hour incubation at 37 °C with a Trypsin/Lys-C protease mixture. After sample clean-up with the Peptide Clean-up Plate, samples were dried under nitrogen at 60 °C, before being resuspended in a 100 μL mixture of 97:3 (v/v) water:acetonitrile, containing 0.1% formic acid.

Samples were kept at 12 °C during analysis and 10 μl of each sample was injected. Injection order for samples was random, with injections of a pooled sample at the start and end, as well as at varying intervals throughout the series. Chromatographic separation was achieved on a Waters™ Acquity UPLC, using a Waters™ Acquity UPLC BEH C18 Column (130 Å, 1.7 μm, 2.1 mm × 50 mm) (186002350), equipped with a Waters™ Acquity UPLC Vanguard BEH C18 precolumn (186003975). Column temperature was held at 60 °C. Mobile phase consisted of (A) water and (B) acetonitrile, both containing 0.1% formic acid. Using a starting flow rate of 0.5 ml/min, the LC gradient consisted of: Dwell at 3% B for 0–0.1 min; Ramp to 40% B at 4.3 min; Ramp to 80% B at 4.31 min with a flow rate of 0.85 ml/min; Dwell at 80% B for 4.31–4.40 min with a flow rate of 0.85 ml/min; Ramp to 3% B at 4.50 min with a flow rate of 0.6 ml/min; Dwell at 3% B for 4.5–5 min with a flow rate of 0.5 ml/min.

MS data were acquired with a Bruker timsTOF Pro 2 using positive ionization in DIA-PASEF mode as previously reported (Szyrwiel et al, 2023). A detailed report of the MS method is provided in the Appendix.

DIA-PASEF data files were processed using DIA-NN version 1.8.1 (Demichev et al, 2020), using an empirically generated spectral library for human proteins provided by Bruker Daltonics. The following DIA-NN options were enabled: Reannotate,

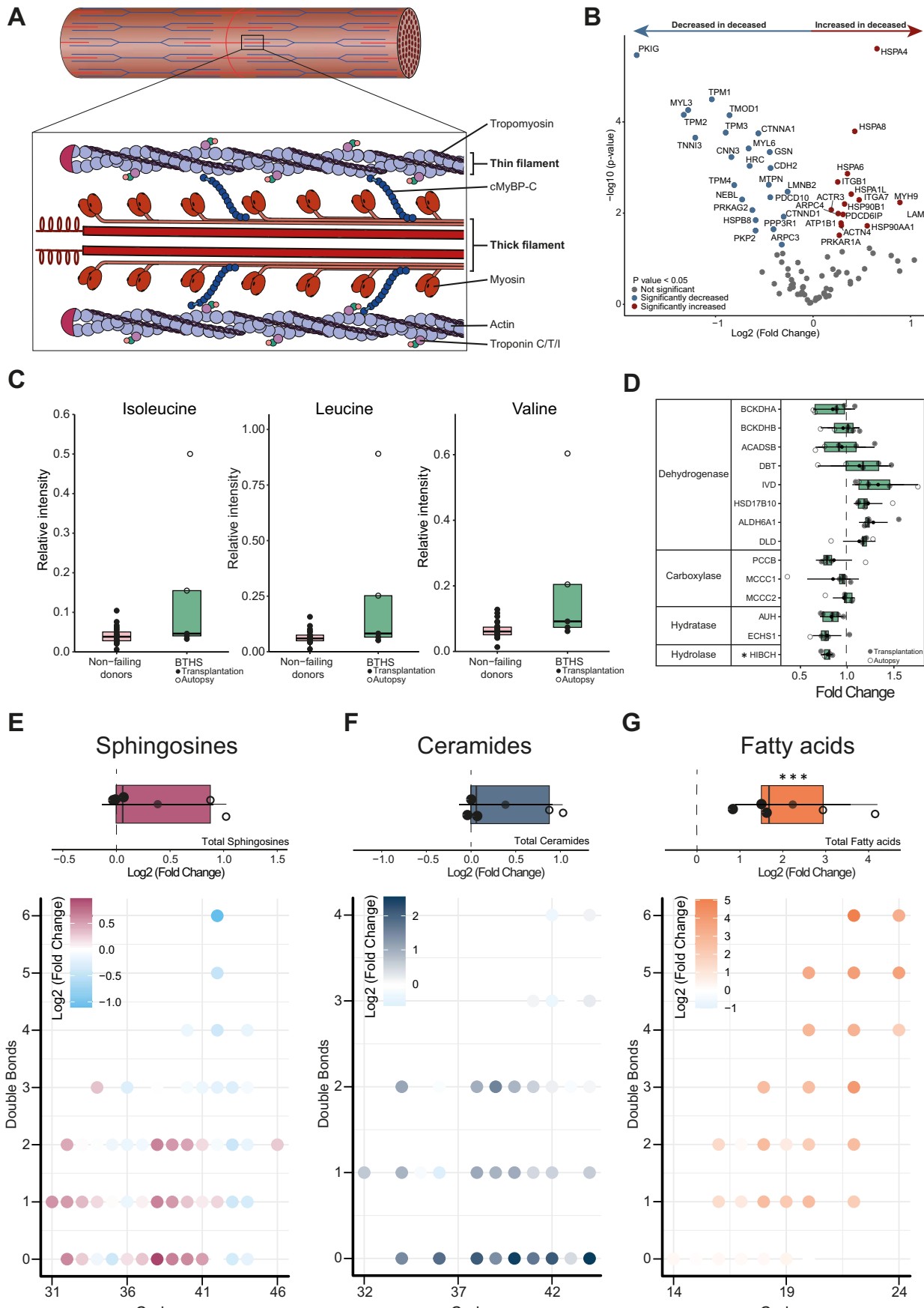

**Figure 5. Cardiac remodelling in BTHS cardiac tissue.**

(A) Structure of cardiac actin-myosin cytoskeleton. (B) Volcano plot of cardiac structural proteins depicting depleted proteins (blue) and accumulated proteins (red) in cardiac tissue of BTHS individuals obtained at autopsy ($n = 2$) and transplantation ($n = 3$). Statistical comparisons were performed using moderated t-tests from the limma package in R, with Benjamini–Hochberg correction. (C) Relative intensity of isoleucine, leucine and valine in both BTHS ($n = 5$) and non-failing donors groups ($n = 24$). Boxplots are defined as in Fig. 2B. Statistical comparisons were performed using moderated t-tests from the limma package in R, with Benjamini–Hochberg correction. (D) Changes in enzymes involved in branched-chain amino acid metabolism in cardiac tissue of BTHS individuals ($n = 5$) when compared to non-failing donors ($n = 24$). Boxplots are defined as in Fig. 2B. Statistical comparisons were performed using moderated t-tests from the limma package in R, with Benjamini–Hochberg correction. (E–G) Changes in total sphingosines, ceramides and fatty acids in cardiac tissue of BTHS individuals ($n = 5$) compared to non-failing donors ($n = 24$). Boxplots are defined as in Fig. 2B. Statistical comparisons were performed using moderated t-tests from the limma package in R, with Benjamini–Hochberg correction. Exact $p$-values corresponding to all statistical comparisons are provided in: Dataset EV4 (STATISTICS LIPIDOMICS), Dataset EV5 (STATISTICS METABOLOMICS), and Dataset EV6 (STATISTICS PROTEOMICS).

## The paper explained

### Problem

Barth syndrome (BTHS) is a rare genetic disorder that severely affects heart function due to mitochondrial abnormalities. Despite its clinical severity, the underlying mechanisms remain poorly understood, limiting treatment options.

### Results

Using a novel integrated multi-omics approach, this study analysed cardiac tissue from BTHS patients and identified profound mitochondrial dysfunction, altered lipid metabolism, and a shift in energy substrate usage. The data also revealed significant inter-individual variability among patients.

### Impact

This comprehensive molecular mapping provides new insights into the cardiac pathology of BTHS and establishes a powerful analytical framework for studying metabolic diseases. It lays the groundwork for future therapeutic strategies and personalized medicine approaches.

Contaminants, N-term M excision, C carbamidomethylation, MBR, No shared spectra. Other settings were as follows: Missed cleavages: 1; Peptide length range: 7–30; Precursor charge range: 1–4; Precursor $m/z$ range: 300–1800; Fragment ion $m/z$ range: 200–1800; Precursor FDR (%): 1; Mass accuracy: 0.0; MS1 accuracy: 0.0; Scan window: 0; Protein names (from FASTA); Double-pass mode; Quant UMS (high precision); RT-dependent; IDs, RT & IM profiling; Optimal results. Data were normalized in DIA-NN using MaxLFQ (Cox et al, 2014).

## Statistics

Statistical analysis and visualization of the acquired data were done in the R programming language (http://ww.r-project.org) using the Limma (Ritchie et al, 2015), ggplot2 (Wickham, 2016), ropls (Thévenot et al, 2015) and mixOmics (Rohart et al, 2017) packages. All comparisons between BTHS affected individuals and non-failing donors were performed using a Bayes moderated t-test using the Limma package, after imputation of missing values using a KNN algorithm. Adjusted $p$-values were subsequently obtained using a Benjamin-Hochberg correction. For other comparisons, for instance within the patient group (i.e. transplant vs autopsy), or comparisons concerning summed analytes (e.g. lipid classes) or grouped analytes (e.g. electron transport chain proteins), methods

are described in the text and figures. In general, analytes containing more than 10% missing values were excluded from analysis, unless the missing values were group-specific.

## Graphics

Graphics in Figs. 2A, 3A and 5A were created with BioRender.com.

## Data availability

The datasets produced in this study are available in the following databases: Proteomics: ProteomeXchange with identifier PXD067204. Metabolomics: MetaboLights with identifier REQ20250715211819 (https://www.ebi.ac.uk/metabolights/editor/MTBLS12953/descriptors). Lipidomics: MetaboLights with identifier REQ20250715211819 (https://www.ebi.ac.uk/metabolights/editor/MTBLS12953/descriptors).

The source data of this paper are collected in the following database record: biostudies:S-SCDT-10_1038-S44321-025-00320-5.

## Peer review information

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

## Acknowledgements

We wish to thank all the individuals affected with BTHS, family members and the staff from all departments that contributed to this study. In particular, we would like to express our sincere gratitude to the Barth Syndrome Foundation for their invaluable assistance. This work is financially supported by Human measurement models 2.0: for health research on disease and prevention, from NWO (no. 18953). SM was supported by an International Postdoc grant from Independent Research Fund Denmark (1057-00039B) and by the Ministry of VWS Sectorplan 'Versnellen op Gezondheid'. RHH was supported by a grant from the Barth Syndrome Foundation.

## Author contributions

**Bauke V Schomakers**: Conceptualization; Data curation; Formal analysis; Investigation; Visualization; Methodology; Writing—original draft; Writing—review and editing. **Adriana S Passadouro**: Conceptualization; Data curation; Formal analysis; Validation; Investigation; Visualization; Writing—original draft; Writing—review and editing. **Maria M Trętowicz**: Data curation; Methodology. **Pelle J Simpson**: Data curation; Formal analysis. **Yorrick R J Jaspers**: Methodology. **Michel van Weeghel**: Methodology. **Iman Man Hu**: Data curation. **Cathelijne M E Lamboo**: Methodology. **Denise Cloutier**: Resources. **Barry J Byrne**: Resources. **Jan Bert van Klinken**: Formal analysis; Validation. **Paul M L Janssen**: Resources. **Sander R Piersma**: Methodology. **Connie R Jimenez**: Methodology. **Frédéric M Vaz**: Resources; Software; Methodology. **Gajja S Salomons**: Supervision; Writing—review and editing. **Jolanda van der Velden**: Conceptualization; Funding acquisition; Writing—review and editing. **Riekelt H Houtkooper**: Conceptualization; Data curation; Supervision; Funding acquisition; Validation; Visualization; Writing—original draft; Writing—review and editing. **Signe Mosegaard**: Conceptualization; Data curation; Supervision; Validation; Investigation; Visualization; Writing—original draft; Project administration; Writing—review and editing.

Source data underlying figure panels in this paper may have individual authorship assigned. Where available, figure panel/source data authorship is listed in the following database record: biostudies:S-SCDT-10_1038-S44321-025-00320-5.

## Disclosure and competing interests statement

The authors declare no competing interests.

# Expanded View Figures

**Figure EV1. Proteomics overview of BTHS ($n = 5$; 2 collected at autopsy, 3 at transplantation) and young male non-failing donors ($n = 5$).**

(A) Heatmap representing z-scores of the top 25 (ranked on *P*-value) proteins. (B) Volcano plot of mitochondrial proteins (according to MitoCarta 3.0), depicting depleted- (blue) and accumulated proteins (red) in cardiac tissue of BTHS individuals ($n = 5$) compared to young male non-failing donors ($n = 5$). Statistical comparisons were performed using moderated t-tests from the limma package in R, with Benjamini–Hochberg correction. Exact *p*-values corresponding to all statistical comparisons are provided in: Dataset EV4 (STATISTICS LIPIDOMICS), Dataset EV5 (STATISTICS METABOLOMICS), and Dataset EV6 (STATISTICS PROTEOMICS).

▶

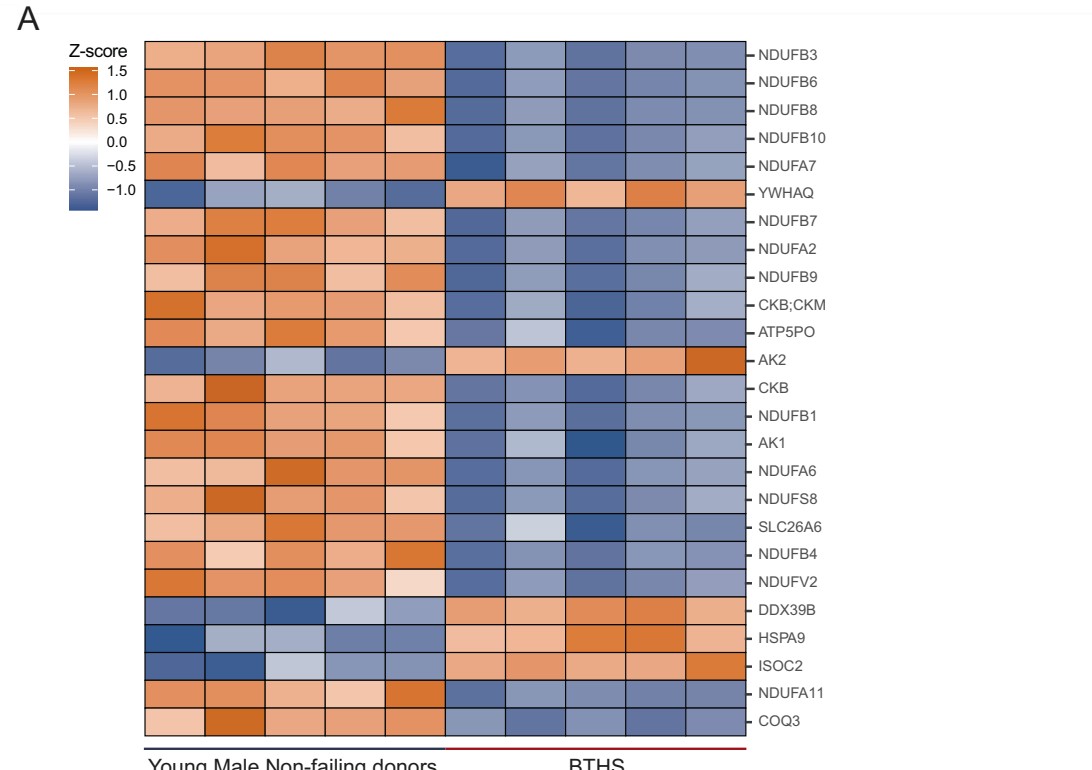

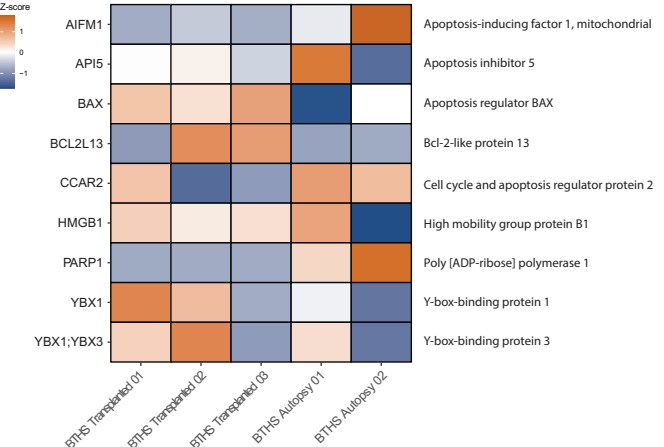

**Figure EV2. Apoptosis/Necrosis in BTHS cardiac tissue.**

Heatmap representing z-scores of an apoptosis/necrosis proteomics subset, alphabetically ordered. BTHS samples ($n = 5$; 2 collected at autopsy, 3 at transplantation) were compared between autopsy and transplantation.

