## [Peer Review File · EMBO Molecular Medicine]

Integrated Multi-Omics Mapping of Mitochondrial Dysfunction and Substrate Preference in Barth Syndrome Cardiac Tissue

Bauke Schomakers, Adriana Passadouro, Maria Trętowicz, Pelle Simpson, Yorrick Jaspers, Michel van Weeghel, Iman Hu, Cathelijne Lamboo, Denise Cloutier, Barry Byrne, Jan Bert van Klinken, Paul Janssens, Sander Piersma, Connie Jimenez, Frédéric Vaz, Gajja Salomons, Jolanda van der Velden, Riekelt Houtkooper, and Signe Mosegaard

Corresponding authors: Signe Mosegaard (s.m.nielsen@amsterdamumc.nl) , Riekelt Houtkooper (r.h.houtkooper@amsterdamumc.nl)

Review Timeline:

Submission Date:	17th Apr 25
Editorial Decision:	19th May 25
Revision Received:	13th Aug 25
Editorial Decision:	2nd Sep 25
Revision Received:	23rd Sep 25
Accepted:	26th Sep 25

Editor: Lise Roth

Transaction Report:

19th May 2025

Dear Dr. Mosegaard,

Thank you for submitting your manuscript to EMBO Molecular Medicine. We have now received feedback from the three reviewers who agreed to evaluate your manuscript. As you will see from the reports below, they acknowledge the interest of the study and support publication of your work, provided that appropriate revisions are made. While the reviewers acknowledge the value of the samples and analyses performed, they also raise concerns relating to confounding variables and comparisons with healthy samples that differ in terms of age and sex.

EMBO Molecular Medicine only allows a single round of revisions, and acceptance or rejection of the manuscript will depend on how complete your responses are in the final version. For this reason, and to save you frustration later on, I would strongly discourage you from submitting an incomplete revision.

We are expecting your revised manuscript within three to four months, if you anticipate any delay, please contact us.

We require:

4) A .docx formatted letter INCLUDING the reviewers' reports and your detailed point-by-point responses to their comments. As part of the EMBO Press transparent editorial process, the point-by-point response is part of the Review Process File (RPF), which will be published alongside your paper.

5) A complete author checklist, which you can download from our author guidelines (<https://www.embopress.org/page/journal/17574684/authorguide#submissionofrevisions>). Please insert information in the checklist that is also reflected in the manuscript. The completed author checklist will also be part of the RPF.

6) All Materials and Methods need to be described in the main text using our 'Structured Methods' format. According to this format, the Methods section includes a Reagents and Tools Table (listing key reagents, experimental models, software and relevant equipment and including their sources and relevant identifiers) followed by a Methods and Protocols section describing the methods, ideally using a step-by-step protocol format. The aim is to facilitate adoption of the methodologies across labs. Please download and fill our Reagents and Tools Table template (.docx), which you can find in our author guidelines:

7) Please note that all corresponding authors are required to supply an ORCID ID for their name upon submission of a revised manuscript.

8) It is mandatory to include a 'Data Availability' section after the Materials and Methods. Before submitting your revision, primary datasets produced in this study need to be deposited in an appropriate public database, and the accession numbers and database listed under 'Data Availability'. Please remember to provide a reviewer password if the datasets are not yet public (see <https://www.embopress.org/page/journal/17574684/authorguide#dataavailability>).

9) For data quantification: please specify the name of the statistical test used to generate error bars and P values, the number (n) of independent experiments (specify technical or biological replicates) underlying each data point and the test used to calculate p-values in each figure legend. The figure legends should contain a basic description of n, P and the test applied. Graphs must include a description of the bars and the error bars (s.d., s.e.m.). Please provide exact p values.

10) Our journal encourages inclusion of *data citations in the reference list* to directly cite datasets that were re-used and obtained from public databases. Data citations in the article text are distinct from normal bibliographical citations and should directly link to the database records from which the data can be accessed. In the main text, data citations are formatted as follows: "Data ref: Smith et al, 2001" or "Data ref: NCBI Sequence Read Archive PRJNA342805, 2017". In the Reference list, data citations must be labeled with "[DATASET]". A data reference must provide the database name, accession number/identifiers and a resolvable link to the landing page from which the data can be accessed at the end of the reference. Further instructions are available at .

11) We replaced Supplementary Information with Expanded View (EV) Figures and Tables that are collapsible/expandable online. EV Figures should be cited as 'Figure EV1, Figure EV2' etc... in the text and their respective legends should be included in the main text after the legends of regular figures.

12) The paper explained: EMBO Molecular Medicine articles are accompanied by a summary of the articles to emphasize the major findings in the paper and their medical implications for the non-specialist reader. Please provide a draft summary of your article highlighting

13) Author contributions: CRediT has replaced the traditional author contributions section because it offers a systematic machine readable author contributions format that allows for more effective research assessment. Please remove the Authors Contributions from the manuscript and use the free text boxes beneath each contributing author's name in our system to add specific details on the author's contribution. More information is available in our guide to authors.

Please also suggest a visual abstract to illustrate your article as a PNG file 550 px wide x 300-600 px high. A cropped portion of this image will serve as thumbnail for the table of content on our webpage.

16) As part of the EMBO Publications transparent editorial process initiative (see our Editorial at <http://embomolmed.embopress.org/content/2/9/329>), EMBO Molecular Medicine will publish online a Review Process File (RPF) to accompany accepted manuscripts.

In the event of acceptance, this file will be published in conjunction with your paper and will include the anonymous referee reports, your point-by-point response and all pertinent correspondence relating to the manuscript. Let us know whether you agree with the publication of the RPF and as here, if you want to remove or not any figures from it prior to publication. Please note that the Authors checklist will be published at the end of the RPF.

EMBO Molecular Medicine has a "scooping protection" policy, whereby similar findings that are published by others during review or revision are not a criterion for rejection. Should you decide to submit a revised version, I do ask that you get in touch

after three months if you have not completed it, to update us on the status.

I look forward to receiving your revised manuscript.

Yours sincerely,

Lise Roth

***** Reviewer's comments *****

Referee #1 (Comments on Novelty/Model System for Author):

The model (cardiac tissue taken from patients) is extremely interesting, relevant and difficult to obtain. Finding disease specific heart defects would be of high value medically. However, the difference in ages between the 5 patients (5 months-15 years) and 24 controls (19-71 years) and the fact that 2 of 5 patient hearts were obtained at autopsy makes determining which changes are "disease specific" nearly impossible.

Referee #1 (Remarks for Author):

The manuscript "Integrated Multi-Omics Mapping of Mitochondrial Dysfunction and Substrate Preference in Barth Syndrome Cardiac Tissue" by Schomakers et al. describes a comparison of multi-omic (metabolomic, lipidomic, proteomic) findings from cardiac samples obtained from 5 individuals with Barth syndrome with 24 control individuals. Cardiomyopathy is an important clinical manifestation of Barth syndrome, and insights into tissue specific changes could be very valuable to better understand this disorder and to inform (new and current) treatments.

It cannot be over-stated how rare and valuable heart samples from patients with Barth syndrome are. The ability to obtain such clinically relevant tissue from individuals with this (or any) ultra-rare disease is normally impossible, so the authors should be applauded for obtaining these samples and doing their best to perform the analyses they feel are most informative. In addition, they have used a very elegant approach which allows all three omics layers to be obtained from a single sample. This not only maximizes tissue utilization, it empowers comparisons not just at each single omics layer, but across all three. Their choice of which omics layers to interrogate is also commendable, and likely more informative than e.g. genomics or transcriptomics might have been. The authors have focused on the likely most relevant metabolic pathways, and have put their findings within the context of known disease mechanisms and other model systems. Overall, it is a very well written manuscript, with a clear approach and without over-statement or over interpretation.

I really like the approach used in this paper, but reading it and looking at the figures, I am just not confident that the findings the authors point to as significant are really driven by disease. If the authors had 10 more patient derived samples and 20 age-matched controls, would these findings really hold up? I'm not sure. And because of that, I think it is up to the authors to convince the reader that these differences are truly disease specific.

With that said, the biggest weakness to the submitted study, as I'm sure the authors are fully aware, are the confounding variables. This study compares only 5 patient derived cardiac samples (3 were obtained at transplant, 2 at autopsy, individuals aged 5 months - 15 years) with 24 non-failing controls (aged 19 years - 71 years). It's a shame the authors did not seem to have access to the age- and sex-matched control tissue previously used by a different group who studied this same patient-derived cardiac tissue (Chatfield et al., J Inherit Metab Dis 2022). Although the authors do a good job of indicating which patient samples were from transplant and which at autopsy, there is at all times three important variables which will affect every result: disease/non-disease, live/dead tissue, and age. The authors are interested only in the first variable, generally display (but do not account for) the second variable, and do not (to my knowledge) attempt to correct for the third variable. Minimally, if they have not already done so (I did not see that they have) the authors need to re-assess and account for age as a co-variate in all of their findings. Maximally, they should look into obtaining and comparing cardiac tissue of Chatfield et al..

Regarding the live/dead tissue - can the authors look for markers of necrosis/apoptosis or similar in the proteomics or metabolomics findings? As the authors correctly point out, some of the differences they find are completely driven by the samples taken at autopsy, and I would be extremely hesitant to interpret these findings as "disease specific".

Finally, it is not clear the total number of proteins, metabolites and lipids were included in each analysis. It would be much more convincing if they found e.g. beta-oxidation enzymes to be changed from a panel 5000 proteins compared to just 50. I would therefore encourage the authors to provide as supplementary information the typical graphs / tables in omics papers of how many proteins, lipids and metabolites were identified in total as well as those which were identified in all samples.

Referee #2 (Remarks for Author):

This study uses multi-omics to analysis the metabolic changes/shifts in Barth Syndrome (BTHS) in cardiac tissues from patients, which is of important clinical implications. However, there are a few concerns here:

1. One of the major concerns is the choose of patient subjects. As also pointed out in the discuss as a limitation that the nonfailing controls are a mix of male and female with various ages. Given that BTHS is a X-linked recessively inherited mitochondrial disorder. It would be better to choose those nonfailing controls that are male. Looking at the list, there are at least 12 samples from male, which would be enough for analysis.
2. Similar to Q1, patients with BTHS mostly develop cardiomyopathy at very early of age. Although it would be hard to have samples from age matched nonfailing controls. It would be better to choose those with relatively young age. This is important that it is well accepted that age is a strong risk factor for heart disease.
3. In Figure 1D-F, it is not clear that for what reason the authors performed analysis with biological triplicates. Is this just to increase the n number of each group?
4. Some discussions about the different patterns seen between metabolomics and lipidomic/proteomcs would be great. In Figure 1D-F, it seems like the difference between nonfailing and BTHS in metabolomics is not that obvious (vs. the comparison in lipidomics and proteomics).
5. Minor: it would be good to show mitochondrial structural changes in nonfailing controls vs those with BTHS.

Dear EMBO editors,

We would like to thank you for considering our manuscript with major revisions and hereby send you our edited manuscript. We have taken the reviewers comments into consideration in the edited manuscript and answer their requests point by point in the text below.

Referee #1:

The manuscript "Integrated Multi-Omics Mapping of Mitochondrial Dysfunction and Substrate Preference in Barth Syndrome Cardiac Tissue" by Schomakers et al. describes a comparison of multi-omic (metabolomic, lipidomic, proteomic) findings from cardiac samples obtained from 5 individuals with Barth syndrome with 24 control individuals. Cardiomyopathy is an important clinical manifestation of Barth syndrome, and insights into tissue specific changes could be very valuable to better understand this disorder and to inform (new and current) treatments.

It cannot be over-stated how rare and valuable heart samples from patients with Barth syndrome are. The ability to obtain such clinically relevant tissue from individuals with this (or any) ultra-rare disease is normally impossible, so the authors should be applauded for obtaining these samples and doing their best to perform the analyses they feel are most informative. In addition, they have used a very elegant approach which allows all three omics layers to be obtained from a single sample. This not only maximizes tissue utilization, it empowers comparisons not just at each single omics layer, but across all three. Their choice of which omics layers to interrogate is also commendable, and likely more informative than e.g. genomics or transcriptomics might have been. The authors have focused on the likely most relevant metabolic pathways, and have put their findings within the context of known disease mechanisms and other model systems. Overall, it is a very well written manuscript, with a clear approach and without over-statement or over interpretation.

I really like the approach used in this paper, but reading it and looking at the figures, I am just not confident that the findings the authors point to as significant are really driven by disease. If the authors had 10 more patient derived samples and 20 age-matched controls, would these findings really hold up? I'm not sure. And because of that, I think it is up to the authors to convince the reader that these differences are truly disease specific.

With that said, the biggest weakness to the submitted study, as I'm sure the authors are fully aware, are the confounding variables. This study compares only 5 patient derived cardiac samples (3 were obtained at transplant, 2 at autopsy, individuals aged 5 months - 15 years) with 24 non-failing controls (aged 19 years - 71 years). It's a shame the authors did not seem to have access to the age- and sex-matched control tissue previously used by a different group who studied this same patient-derived cardiac tissue (Chatfield et al., J Inherit Metab Dis 2022). Although the authors do a good job of indicating which patient samples were from transplant and which at autopsy, there is at all times three important variables which will affect every result: disease/non-disease, live/dead tissue, and age. The authors are interested only in the first variable, generally display (but do not account for) the second variable, and do not (to my knowledge) attempt to correct for the third variable. Minimally, if they have not already done so (I did not see that they have) the authors need to re-assess and account for age as a co-variate in all of their findings. Maximally, they should look into obtaining and comparing cardiac tissue of Chatfield et al..

We thank the reviewer for their kind words and thorough evaluation of our work. We acknowledge the limitations of our control group. We agree that age-matched male donors such as those in Chatfield et al. would have been the optimal choice, unfortunately, such group was not available for us when performing this study.

To address the reviewer's concern, we performed additional statistical analyses focusing exclusively on the five youngest male donors from the control group. These analyses reaffirmed the robustness

and disease specificity of our main findings, particularly those related to lipidome alterations and mitochondrial dysfunction. The proteomics dataset forms the foundation of our study, underpinning the observed metabolic and lipidomic differences. To further validate our results, we compared the top 25 most altered proteins between the youngest male non-failing donors and BTHS samples (Figure Expanded View (EV)1A). We also include a volcano plot of all mitochondrial proteins for the same comparison (Figure EV1B). Both analyses show results that are almost identical to those obtained using the full control group, reinforcing the conclusion that our findings are disease-specific rather than driven by potential confounders.

That said, due to their high-dimensional nature, omics analyses benefit significantly from larger sample sizes. The smaller dataset might alleviate one possible confounder, but the exclusion of a significant part of the samples will make the analyses more vulnerable to others. This is particularly relevant for human-derived samples such as ours, collected in real-world settings, where inter-individual variability is inherently high. Our current control group is well-balanced in terms of sex, age, and BMI, providing a broad representation of human biological diversity. This balance reduces the likelihood that any single demographic factor disproportionately influences the results. Using the full control set therefore increases, rather than decreases, the likelihood that observed differences are disease-specific and not caused by demographic or environmental confounders.

Nonetheless, we recognize this as a limitation and have addressed it in two ways:

1. We provide full sample metadata and raw data to enable further stratified analyses by interested readers.
2. We have been cautious in our interpretations throughout the manuscript, avoiding overstatements or potentially controversial claims. We instead present a uniquely comprehensive and transparent overview of these highly rare samples, while making sure any conclusions are strongly anchored in existing literature.

Specifically, we have added this statement to highlight the additional analyses and potential confounders (Page 11, line 300-307):

“Finally, to ensure that the observed differences in our study between non-failing donors and BTHS are disease-specific and not the result of potential confounders such as sex or age, we repeated the proteomics analysis that forms the core of most of our results, using only the five youngest male non-failing donors (Figure EV1A). The top altered proteins show a highly similar signature as when compared to our total control group, highlighting that the observed differences are likely not affected by the varied differences of sex and age in our control group, but are true disease-specific differences in the BTHS heart.”

As well as this statement (Page 12, line 332-335):

“We repeated the analysis using only the youngest male non-failing donors (n=5) to provide additional confidence that the observed changes are disease-specific (Figure EV1B). The proteomic alterations remained highly consistent, underscoring the robustness of these changes in BTHS hearts.”

Regarding the live/dead tissue - can the authors look for markers of necrosis/apoptosis or similar in the proteomics or metabolomics findings? As the authors correctly point out, some of the differences they find are completely driven by the samples taken at autopsy, and I would be extremely hesitant to interpret these findings as "disease specific".

Indeed, we agree that within our study, the analysis of the hearts taken at autopsy vs those taken during transplantation is the most sensitive to confounding factors. This stems from both the limited sample size and the clear differences in sampling conditions. As such, the final figure in the paper is our most speculative and we welcome this reviewer's suggestions to further explore and improve this part of the manuscript. To address this, we compiled a panel of proteins associated with apoptosis and necrosis based on a literature search, including: AIFM1, API5, BAX, BCL2L13,

CCAR2, HMGB1, PARP1, YBX1 and YBX3. These markers were selected based on their established roles in cell death pathways and availability in our proteomics dataset. However, careful analysis did not reveal any pattern indicative of increased apoptotic or necrotic activity in either group.

Despite this, we share the reviewer's caution and have taken care to present this part of the study as strictly exploratory. We have clearly stated the limitations of this analysis in the manuscript and have refrained from making any statistical claims or drawing definitive conclusions from this data. Our intention is to transparently share potentially valuable observations while acknowledging the inherent constraints of the dataset.

We have amended our manuscript to accurately address these considerations with the following statement (Page 14, line 380-383):

"As this analysis can be sensitive to confounders due to the low sample numbers and inherent sampling differences, we also investigated if the failing hearts showed a more generic apoptotic/necrotic phenotype using a subset of established protein markers. No distinct pattern was observed (Figure EV2A)."

Finally, it is not clear the total number of proteins, metabolites and lipids were included in each analysis. It would be much more convincing if they found e.g. beta-oxidation enzymes to be changed from a panel 5000 proteins compared to just 50. I would therefore encourage the authors to provide as supplementary information the typical graphs / tables in omics papers of how many proteins, lipids and metabolites were identified in total as well as those which were identified in all samples.

We fully agree that transparency is essential in omics analyses to avoid bias and overly optimistic interpretations. In line with this, our data presentation is designed to be comprehensive: we consistently display all relevant proteins or metabolites, not just those that meet significance thresholds. For example, in Figure 4, we include all mitochondrial beta-oxidation enzymes identified in our dataset, regardless of whether they show significant changes. This approach ensures that readers have access to the full context and can independently assess the biological relevance of observed patterns. This is in line with our general goal of providing a comprehensive overview of these samples, rather than make large or novel claims. To support our broader aim of transparency, we include all three complete datasets for further analysis, and have amended our manuscript to indicate total numbers of identified metabolites, lipids, and proteins (Page 10, line 256–257). These are 123 metabolites, 1899 lipids and 1445 proteins.

Referee #2:

This study uses multi-omics to analysis the metabolic changes/shifts in Barth Syndrome (BTHS) in cardiac tissues from patients, which is of important clinical implications. However, there are a few concerns here:

1. One of the major concerns is the choose of patient subjects. As also pointed out in the discuss as a limitation that the nonfailing controls are a mix of male and female with various ages. Given that BTHS is a X-linked recessively inherited mitochondrial disorder. It would be better to choose those nonfailing controls that are male. Looking at the list, there are at least 12 samples from male, which would be enough for analysis.

We appreciate the reviewer's thoughtful and engaging comments and fully acknowledge the limitations of our control group, both here and in the manuscript. Indeed, age-matched male donors would be the most appropriate controls for a study of BTHS, given its X-linked inheritance.

To address this concern, we performed additional statistical analyses focusing exclusively on the five youngest male donors. These analyses reaffirmed the robustness and disease specificity of our main findings, particularly those related to lipidome alterations and mitochondrial dysfunction. The proteomics dataset forms the foundation of our study, underpinning the observed metabolic and lipidomic differences. To further validate our results, we compared the top 25 most altered proteins between the youngest male non-failing donors and BTHS samples (Figure EV1A). We also include a volcano plot of all mitochondrial proteins for the same comparison (Figure EV1B). Both analyses show results that are almost identical to those obtained using the full control group, reinforcing the conclusion that our findings are disease-specific rather than driven by potential confounders.

That said, due to their high-dimensional nature, omics analyses benefit significantly from larger sample sizes. The smaller dataset might alleviate one possible confounder, but the exclusion of a significant part of the samples will make the analyses more vulnerable to others. This is particularly relevant for human-derived samples such as ours, collected in real-world settings, where inter-individual variability is inherently high. Our current control group is well-balanced in terms of sex, age, and BMI, providing a broad representation of human biological diversity. This balance reduces the likelihood that any single demographic factor disproportionately influences the results. Using the full control set therefore increases, rather than decreases, the likelihood that observed differences are disease-specific and not caused by demographic or environmental confounders.

Nonetheless, we recognize this as a limitation and have addressed it in two ways:

1. We provide full sample metadata and raw data to enable further stratified analyses by interested readers.
2. We have been cautious in our interpretations throughout the manuscript, avoiding overstatements or potentially controversial claims. We instead present a uniquely comprehensive and transparent overview of these highly rare samples, while making sure any conclusions are strongly anchored in existing literature.

Specifically, we have added this statement to highlight the additional analyses and potential confounders (Page 11, line 300-307)

Finally, to ensure that the observed differences in our study between non-failing donors and BTHS are disease-specific and not the result of potential confounders such as sex or age, we repeated the proteomics analysis that forms the core of most of our results, using only the five youngest male non-failing donors (Figure EV1A). The top altered proteins show a highly similar signature as when compared to our total control group, highlighting that the observed differences are likely not affected by the varied differences of sex and age in our control group, but are true disease-specific differences in the BTHS heart.

As well as this statement (Page 12, line 332-335):

“We repeated the analysis using only the youngest male non-failing donors (n=5) to provide additional confidence that the observed changes are disease-specific (Figure EV1B). The proteomic alterations remained highly consistent, underscoring the robustness of these changes in BTHS hearts.”

2. Similar to Q1, patients with BTHS mostly develop cardiomyopathy at very early of age. Although it would be hard to have samples from age matched nonfailing controls. It would be better to choose those with relatively young age. This is important that it is well accepted that age is a strong risk factor for heart disease.:

We kindly refer the reviewer to our answer to the previous comment.

3. In Figure 1D-F, it is not clear that for what reason the authors performed analysis with biological triplicates. Is this just to increase the n number of each group?

BTHS samples, and especially heart tissues, are exceedingly rare, resulting in a limited sample size for our patient group. Additionally, within that modest sample group, there are confounding factors affecting their homogeneity, as this reviewer rightfully points out. To address this, we included biological triplicates in Figures 1A–F not to artificially increase the n number, but to demonstrate the reproducibility and robustness of our analytical method. The minimal variation observed among triplicates underscores the reliability of our approach and provides confidence that any observed variation between BTHS samples is biological in nature, rather than analytical. Importantly, we have not used these triplicates to inflate the n number of BTHS samples, or used them to obtain significance. Instead, all statistics were performed on the averages of the triplicates, compared to non-failing donors. We have amended our manuscript with the following statements to better reflect these choices:

Page 10, line 260-261, concerning the PCA plots in Figure 1A-C:

In all three omics analyses, the tight clustering of these triplicates underscores the robustness of the analytical method we present here.

Page 10, line 270-273, concerning the heatmaps in Figure 1D-F:

This approach allows us to further confirm the robustness of our method by demonstrating high repeatability at the analyte level. Importantly, for all subsequent statistical analyses, we use the average of each triplicate to avoid artificially inflating the sample size.

Page 15, line 404-406, in the discussion:

This approach allowed us to achieve a comprehensive molecular and metabolic characterization of BTHS cardiac tissue with a high degree of certainty, illustrated by the robustness of our BTHS triplicate analysis.

4. Some discussions about the different patterns seen between metabolomics and lipidomic/proteomics would be great. In Figure 1D-F, it seems like the difference between nonfailing and BTHS in metabolomics is not that obvious (vs. the comparison in lipidomics and proteomics).

We indeed recognize this pattern in the data and have amended our manuscript to discuss it in more detail in the discussion (Page 15, line 407-420):

“Notably, within the BTHS patient group, inter-individual differences detected by our metabolomics analysis are larger than those found in our lipidomics or proteomics. This is consistent with our expectations. The primary defect in BTHS lies within the lipid metabolism (specifically, cardiolipin biosynthesis) which leads to a relatively uniform and pronounced alteration in the lipidome across patients. Given cardiolipins central role in maintaining mitochondrial membrane structure, it is not surprising that proteins dependent on membrane integrity, such as those in the electron transport chain, are consistently affected, as reflected in our proteomics data and previous research¹⁵. In contrast, the polar metabolome is further downstream from the primary defect and is influenced by a broader range of factors, including individual compensatory mechanisms and natural metabolic variability. This results in a more heterogeneous metabolic profile among BTHS patients. Nevertheless, we still observe a distinct metabolic signature for BTHS patients when compared to non-failing donors, for instance in pathways related to energy metabolism, which is consistent with other studies, clinical observations, and our other data.”

5. Minor: it would be good to show mitochondrial structural changes in nonfailing controls vs those with BTHS.

We agree that visualizing mitochondrial structural changes in BTHS heart tissues would be highly informative, especially given that such alterations are well-documented in the literature and are strongly suggested by our molecular data. For the current analysis, the heart tissues are required to be freeze-dried, followed by mechanical homogenization. Structural information is unfortunately lost during this process. However, it is well established that mitochondrial morphology is changed in BTHS patients. This was already established in the first literature published by prof. Peter Barth in 1983 (Barth PG, Scholte HR, Berden JA, Van der Klei-Van Moorsel JM, Luyt-Houwen IE, Van 't Veer-Korthof ET, Van der Harten JJ, Sobotka-Plojhar MA. An X-linked mitochondrial disease affecting cardiac muscle, skeletal muscle and neutrophil leucocytes. J Neurol Sci. 1983 Dec;62(1-3):327-55. doi: 10.1016/0022-510x(83)90209-5. PMID: 6142097) and supported by several other studies later.

2nd Sep 2025

Dear Dr. Mosegaard,

Thank you for submitting your revised study. We have now received the reports from the referees. As you will see below, they are satisfied with the revisions, and I will therefore be able to accept your manuscript once the following editorial concerns are addressed:

1/ Manuscript text:

- Please indicate in track changes mode any new modification in the text.

- Please provide up to 5 keywords.

- Please correct the order and headings of the sections to:

Abstract / Keywords / The Paper Explained / Introduction / Results / Discussion / Methods / Data Availability /

Acknowledgements / Disclosure and Competing Interests Statement / References / Figure Legends / Expanded View Figure Legends.

- Methods:

- o Please download and fill our Reagents and Tools Table template (.docx), which you can find in our author guidelines:

<https://www.embopress.org/page/journal/14693178/authorguide#structuredmethods>. When submitting your revised manuscript, please do not include the Reagents and Tools Table in the Methods section of the manuscript but upload it as a separate file choosing the file type "Reagent Table".

- o Please provide a full statement confirming that informed consent was obtained from all subjects and that the experiments conformed to the principles set out in the WMA Declaration of Helsinki and the Department of Health and Human Services Belmont Report.

- o Please state details of authority granting ethics approval (IRB or equivalent committee(s), provide reference number for approval.

- o Statistics: please provide a statement on sample size, inclusion/exclusion criteria, blinding and randomization.

- o Please remove the references to BioRender from the figure legends and add a dedicated section to the Methods using this format:

Graphics:

(some of the... OR Figure #... OR synopsis) Graphics were created with BioRender.com.

- o Please remove the data availability section currently placed after the statistics section.

- Data availability section: please provide direct links to the deposited datasets.

- Please provide a 'disclosure statement and competing interests' statement. See our policy at

<https://www.embopress.org/competing-interests>.

- References: please reformat to have 10 author names listed before et al; DOIs should be removed.

2/ Figures:

- Panel "A" can be removed from Fig EV2 since there is only one panel.

- Suppl. Tables 2 - 4 should be renamed Dataset EV1 - EV3, and each dataset should have a legend added to the file in a separate tab/worksheet.

- Appendix: please add a cover page with a title and table of contents to the supplementary methods and upload as a PDF.

Please also add Supplementary Table 1, which should then be renamed "Appendix Table S1". A legend should be added with a short description of the table.

- Please address the queries from our data editors in the figure legends:

1. Please define the annotated p values ****/****/**/* as well as provide the exact p-values for the same in the legend of figure 2B, D; 4C as appropriate.

2. Please note that the exact p values are not provided in the legends of figures 3B, D, E, G; 4A, B

3. Please indicate the statistical test used for data analysis in the legends of figures 1I, 2B, C, D; 3B-G; 4A-C; 5B, EV1 B

4. Please note that the box plots need to be defined in terms of minima, maxima, centre, bounds of box and whiskers, and percentile in the legends of figures 2B, D; 3B, E, F; 4C, 5C, D, E, F, G

5. Please note that information related to n is missing in the legends of figures 2B, C, D; 3B, C, E, F; 4A-C; 5B-G; EV1 B

3/ Source Data: I understand that you did not upload the individual source data for your figures, since they are all supported by the large-scale data that has been deposited. Please let me know if this is incorrect.

4/ Checklist:

Please fill in the section Ethics - subsection "authority granting ethics approval"

5/ The paper explained: EMBO Molecular Medicine articles are accompanied by a summary of the articles to emphasize the major findings in the paper and their medical implications for the non-specialist reader. Please provide a draft summary of your article highlighting:

6/ Synopsis: Thank you for providing a nice visual abstract. I have cropped a small portion to serve as thumbnail for the table of content on our webpage (attached). Please let us know if you agree, or provide an alternative image at the same dimensions. Please also provide a synopsis text. It should include a short stand first (maximum of 300 characters, including space) as well as 2-5 one-sentence bullet points that summarize the paper (maximum of 30 words / bullet point). They should be designed to be complementary to the abstract - i.e. not repeat the same text.

7/ As part of the EMBO Publications transparent editorial process initiative (see our Editorial at <http://embomolmed.embopress.org/content/2/9/329>), EMBO Molecular Medicine will publish online a Review Process File (RPF) to accompany accepted manuscripts.

This file will be published in conjunction with your paper and will include the anonymous referee reports, your point-by-point response and all pertinent correspondence relating to the manuscript. Let us know whether you agree with the publication of the RPF.

I look forward to receiving your revised manuscript.

Yours sincerely,

Lise Roth

***** Reviewer's comments *****

Referee #1 (Remarks for Author):

Thank you to the authors for doing the additional analysis. The extended tables and figures are much appreciated and give necessary transparency. Thanks also for clarifying the number of proteins, lipids and metabolites covered. It seems that missing values were not inferred (which is good). Although the autopsy samples are often clear outliers, this does not seem to be driven by the detected apoptosis/necrosis markers.

Overall the authors have adequately addressed all of my concerns. Although there are many caveats (well described in the Discussion), this is valuable, interesting work and will be appreciated by the community. Congratulations to the authors on this study.

Referee #2 (Remarks for Author):

Thanks for addressing the comments.

The authors addresses the remaining editorial issues.

26th Sep 2025

Dear Dr. Mosegaard,

Thank you for submitting your revised files. I am pleased to inform you that your manuscript is accepted for publication and is now being sent to our publisher to be included in the next available issue of EMBO Molecular Medicine.

Please note that I introduced minor changes in your manuscript text:

- typos corrected in the last sentence of the abstract
- the ethics paragraph was removed from the Data availability section, and moved to the "Patient samples" section.

Let us know immediately if you don't agree with these changes.

With kind regards,

Lise Roth
